# 🎵 HarmoniCa: Harmonizing Training and Inference for Better Feature Caching in Diffusion Transformer Acceleration

**Yushi Huang**[1 2 * †]  **Zining Wang**[2 3 * †]  **Ruihao Gong**[2 3]  **Jing Liu**[4]  **Xinjie Zhang**[1]  **Jinyang Guo**[3 5]
**Xianglong Liu**[3 6]  **Jun Zhang**[1]

## Abstract

Diffusion Transformers (DiTs) excel in generative tasks but face practical deployment challenges due to high inference costs. Feature caching, which stores and retrieves redundant computations, offers the potential for acceleration. Existing learning-based caching, though adaptive, overlooks the impact of the prior timestep. It also suffers from misaligned objectives–*aligned predicted noise vs. high-quality images*–between training and inference. These two discrepancies compromise both performance and efficiency. To this end, we _harmonize_ training and inference with a novel learning-based _caching_ framework dubbed **HarmoniCa**. It first incorporates *Step-Wise Denoising Training* (SDT) to ensure the continuity of the denoising process, where prior steps can be leveraged. In addition, an *Image Error Proxy-Guided Objective* (IEPO) is applied to balance image quality against cache utilization through an efficient proxy to approximate the image error. Extensive experiments across 8 models, 4 samplers, and resolutions from $256 \times 256$ to $2K$ demonstrate superior performance and speedup of our framework. For instance, it achieves over $40\%$ latency reduction (*i.e.*, $2.07\times$ theoretical speedup) and improved performance on PIXART-$\alpha$. Remarkably, our *image-free* approach reduces training time by $25\%$ compared with the previous method. Our code is available at https://github.com/ModelTC/HarmoniCa.

---

[*]Equal contribution [†]Work done during internship at SenseTime Resaerch [1]iComAI Lab, Hong Kong University of Science and Technology [2]SenseTime Research [3]SKLCCSE, Beihang University [4]ZIP Lab, Monash University [5]IAI, Beihang University [6]SCSE, Beihang University. Correspondence to: Ruihao Gong <gongruihao@buaa.edu.cn>, Jun Zhang <eejzhang@ust.hk>.

*Proceedings of the 42$^{nd}$ International Conference on Machine Learning*, Vancouver, Canada. PMLR 267, 2025. Copyright 2025 by the author(s).

*"A dreamy pastel illustration of a flower-filled meadow beneath drifting clouds, bright blossoms in the foreground and a distant cottage perched on a gentle slope, no humans visible."*

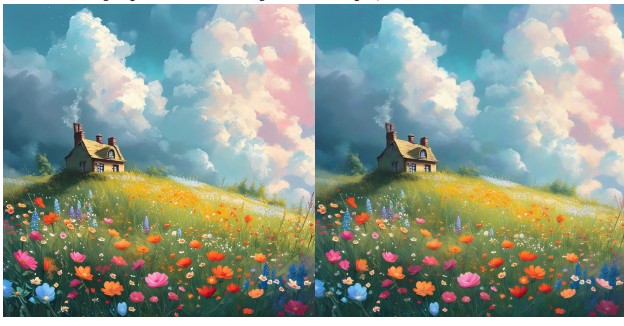

(a) PIXART-$\Sigma$                    (b) HarmoniCa ($1.73\times$)

*Figure 1.* High-resolution $2048 \times 2048$ images generated using PIXART-$\Sigma$ (Chen et al., 2024a) with a 20-step DPM-Solver++ sampler (Lu et al., 2022b). Our proposed feature caching framework achieves a substantial $1.73\times$ speedup with *minimal visual difference*. More visualization results can be found in Sec. O.

## 1. Introduction

Diffusion models (Ho et al., 2020; Dhariwal & Nichol, 2021) have recently gained increasing popularity in a variety of generative tasks, such as image (Saharia et al., 2022; Esser et al., 2024) and video generation (Blattmann et al., 2023; Ma et al., 2024c), due to their ability to produce diverse and high-quality samples. Among different backbones, Diffusion Transformers (DiTs) (Peebles & Xie, 2023) stand out for offering exceptional scalability. However, the extensive parameter size and multi-round denoising nature of diffusion models bring tremendous computational overhead during inference, limiting their practical applications. For instance, generating one $2048 \times 2048$ resolution image using PIXART-$\Sigma$ (Chen et al., 2024a) with 0.6B parameters and 20 denoising rounds can take up to 14 seconds on a single NVIDIA H800 80GB GPU, which is unacceptable.

In light of the aforementioned problem, previous methods emerge from two perspectives: reducing the number of sampling steps (Liu et al., 2022a; Song et al., 2020b) and decreasing the network complexity in noise prediction of each step (Fang et al., 2023; He et al., 2024). Recently, a new

branch of research (Selvaraju et al., 2024; Yuan et al., 2024; Chen et al., 2024b) has started to focus on accelerating sampling time per step by the feature caching mechanism. This technique takes advantage of the repetitive computations across timesteps in diffusion models, allowing previously computed features to be cached and reused in later steps. Nevertheless, most existing methods are either tailored to the U-Net architecture (Ma et al., 2024b; Wimbauer et al., 2024) or develop their strategy based on empirical observations (Chen et al., 2024b; Selvaraju et al., 2024). Therefore, there is a lack of adaptive and systematic approaches for DiT models. Learning-to-Cache (Ma et al., 2024a) introduces a learnable router to guide the caching scheme for DiT models. However, we have found that this method induces discrepancies between training and inference, which always leads to distortion build-up (Ning et al., 2023; Li et al., 2024b; Ning et al., 2024). The discrepancies arise from two main factors: (1) *Prior Timestep Disregard*: During training, the model directly samples a timestep and employs the training images with manually added noise akin to DDPM (Hu et al., 2021). This pattern ignores the impact of the caching mechanism from earlier steps, which differs from the inference process. (2) *Objective Mismatch*: The training objective minimizes noise prediction error of each timestep. Differently, the inference goal aims for high-quality final images, causing a misalignment in objectives. We believe these inconsistencies hinder effective and efficient router learning.

To alleviate the above discrepancies effectively, we harmonize training and inference with HarmoniCa, a novel cache learning framework featuring a unique training paradigm and a distinct learning objective. Specifically, to mitigate the first disparity, we design *Step-Wise Denoising Training* (SDT). It aligns the training process with the full denoising trajectory of inference using a student-teacher model setup. The student utilizes the cache while the teacher does not, effectively mimicking the teacher's outputs across all continuous timesteps. This approach maintains the reuse and update of the cache at earlier timesteps, similar to inference. Additionally, to address the misalignment in optimization goals, we introduce the *Image Error Proxy-Guided Objective* (IEPO). This objective leverages a proxy to approximate the final image error and reduces the significant costs of directly utilizing the error to supervise training. IEPO helps SDT efficiently balance cache usage and image quality. By combining the two techniques, extensive experiments show the promising performance and speedup of HarmoniCa, *e.g.*, over $1.69\times$ speedup and much lower FID (Nash et al., 2021) for non-accelerated PIXART-$\alpha$ (Chen et al., 2023). In addition, HarmoniCa eliminates the requirement of training with a large number of images and reduces about $25\%$ training time compared to the existing learning-based method (Ma et al., 2024a), further enhancing its applicability.

Our contributions are summarized as follows:

- We identify two key discrepancies in existing learning-based caching methods: (1) *Prior Timestep Disregard*, where training neglects the impact of earlier timesteps, and (2) *Objective Mismatch*, which minimizes intermediate output errors rather than final image errors. These issues hinder performance and acceleration improvements.

- We propose HarmoniCa, a framework that resolves these discrepancies by (1) *Step-Wise Denoising Training* (SDT), which captures the full denoising trajectory to account for earlier timesteps, and (2) *Image Error Proxy-Guided Optimization Objective* (IEPO), which aligns the training objective with inference by approximating image error.

- Extensive experiments on NVIDIA H800 80GB GPUs with 8 models, 4 samplers, and 4 resolutions demonstrate the efficacy and universality of our framework. For example, it achieves a $6.74$ IS increase and $1.24$ FID reduction on DiT-XL/2, surpassing previous state-of-the-art (SOTA) methods with a higher speedup ratio. Moreover, our *image-free* framework offers greater efficiency and applicability at significantly lower training costs.

## 2. Related Work

**Diffusion models.** Diffusion models, initially conceptualized with the U-Net architecture (Ronneberger et al., 2015), have achieved satisfactory performance in image (Rombach et al., 2022; Podell et al., 2023) and video generation (Ho et al., 2022). Despite their success, U-Net models struggle with modeling long-range dependencies in complex, high-dimensional data. In response, the Diffusion Transformer (DiT) (Peebles & Xie, 2023; Chen et al., 2023; 2024a) is introduced, leveraging the inherent scalability of Transformers to efficiently enhance model capacities and handle more complex tasks with improved performance.

**Efficent diffusion.** Diverse methods have been proposed to reduce the generation overhead for diffusion models. These techniques fall into two main categories: reducing the number of sampling steps and decreasing the computational load per denoising step. In the first category, several works utilize distillation (Salimans & Ho, 2022; Luhman & Luhman, 2021) to obtain reduced sampling iterations. Furthermore, this category encompasses advanced techniques such as implicit samplers (Kong & Ping, 2021; Song et al., 2020a; Zhang et al., 2022) and specialized differential equation (DE) solvers. These solvers tackle both stochastic differential equations (SDE) (Song et al., 2020b; Jolicoeur-Martineau et al., 2021) and ordinary differential equations (ODE) (Lu et al., 2022a; Liu et al., 2022a; Zhang & Chen, 2022), addressing diverse aspects of diffusion model optimization. In contrast, the second category mainly focuses on model compression (He et al., 2025; Yang et al., 2024; Guo et al., 2024). It leverages techniques like pruning (Guo

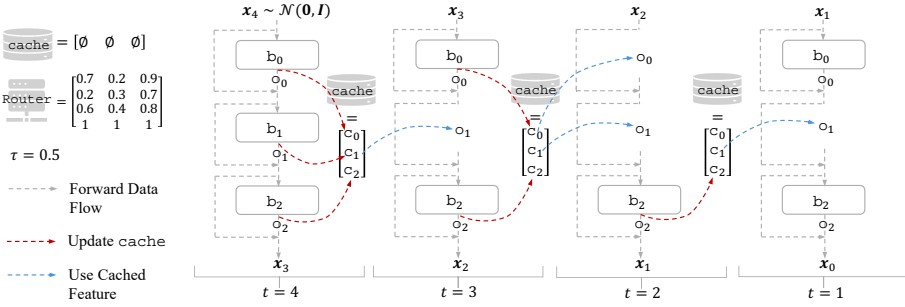

*Figure 2.* Generation process from a random Gaussian noise $x_4$ to an image $x_0$ using feature caching ($T = 4$, $N = 3$). We omit the sampler and conditional inputs.

*Figure 3.* One training iteration of Learning-to-Cache. This method uses the noisy image $x_t$ as the input at $t$. $\mathcal{L}_{LTC}^{(t)}$ denotes the loss function. "$*$" in "DiT ($*$)" represents the timestep.

et al., 2020; Zhang et al., 2024; Wang et al., 2024b;c) and quantization (Shang et al., 2023; Huang et al., 2024a; He et al., 2024; Huang et al., 2024b; Lv et al., 2024) to reduce the workload in a static way. Additionally, dynamic inference compression is also being explored (Liu et al., 2023; Pan et al., 2024), where different models are employed at varying steps of the process. In this work, we focus on the urgently needed DiT acceleration through feature caching, a method distinct from the above-discussed ones.

**Feature caching.** Due to the high similarity between activations (Li et al., 2023b; Wimbauer et al., 2024) across continuous denoising steps in diffusion models, recent studies (Ma et al., 2024b; Wimbauer et al., 2024; Li et al., 2023a) have explored caching these features for reuse in subsequent steps to avoid redundant computations. Notably, their strategies rely heavily on the specialized structure of U-Net, *e.g.*, up-sampling blocks (Ma et al., 2024b) or `SpatialTransformer` blocks (Li et al., 2023a). Besides, FORA (Selvaraju et al., 2024) and Δ-DiT (Chen et al., 2024b) further apply the feature caching mechanism to DiT. However, both methods select the cache position and lifespan in a handcrafted way. Learning-to-Cache (Ma et al., 2024a) introduces a learnable cache scheme but induces discrepancies between training and inference. In this work [1], we design a new learning-based framework to alleviate the discrepancies between the training and inference, which further enhances the performance and acceleration for DiT.

In addition, *token caching* (Zou et al., 2024; Lou et al., 2024), a granular way to reduce computation, recently emerged. It can be seen as an extremely fine-grained feature caching. Although compatible with our work, we only focus on feature caching with *block-wise* granularity as below.

[1]It is worth noting that we focus on the real-time speedup ratio in this paper instead of the theoretical upper bound in some existing works (Zou et al., 2024; Chen et al., 2024b).

## 3. Prelimilaries

**Caching granularity.** The noise estimation network of DiT (Peebles & Xie, 2023) is built on the Transformer block (Vaswani, 2017), which is composed of an Attention block and a feed-forward network (FFN). Each Attention block and FFN is wrapped up in a residual connection (He et al., 2016). For convenience, we sequentially denote these Attention blocks and FFNs without residual connections as $\{b_0, b_1, \ldots, b_{N-1}\}$, where $N$ is their total amount. Following the existing study (Ma et al., 2024a), we store the output of $b_i$ in cache as $c_i$. The cache, once completely filled, is represented as:

$$\texttt{cache} = [c_0, c_1, \ldots, c_{N-1}]. \quad (1)$$

**Caching router.** The caching scheme for DiT can be formulated with a pre-defined threshold $\tau$ ($0 \leq \tau < 1$) and a customized router matrix:

$$\texttt{Router} = [r_{t,i}]_{1 \leq t \leq T, 0 \leq i \leq N-1} \in \mathbb{R}^{T \times N}, \quad (2)$$

where $0 < r_{t,i} \leq 1$ and $T$ is the maximum denoising step. At timestep $t$ during inference, the residual branch corresponding to $b_i$ is fused with $o_i$ defined as follows:

$$o_i = \begin{cases} b_i(\mathbf{h}_i, \mathbf{cs}), & r_{t,i} > \tau \\ c_i, & r_{t,i} \leq \tau \end{cases}, \quad (3)$$

where $\mathbf{h}_i$ is the image feature and $\mathbf{cs}$ represents the conditional inputs [2]. Specifically, $r_{t,i} > \tau$ indicates computing $b_i(\mathbf{h}_i, \mathbf{cs})$ as $o_i$. This computed output also replaces $c_i$ in the `cache`. Otherwise, the model loads $c_i$ from `cache` without computation. A naive example of the caching scheme is depicted in Fig. 2. To be noted, $\texttt{Router}_{T,:}$ is set to $[1]_{1 \times N}$ by default to pre-fill the empty `cache`.

**Cache usage ratio (CUR).** In addition, we define cache usage ratio (CUR) formulated as $\frac{\sum_{t=1}^{t=T} \sum_{i=0}^{N-1} \mathbb{I}_{r_{t,i} \leq \tau}}{N \times T}$ in this paper to represent the reduced computation from reusing cached features. In Fig. 2, CUR is roughly equal to 33.33%.

[2]For example, $\mathbf{cs}$ represents the time condition and textual condition for text-to-image (T2I) generation.

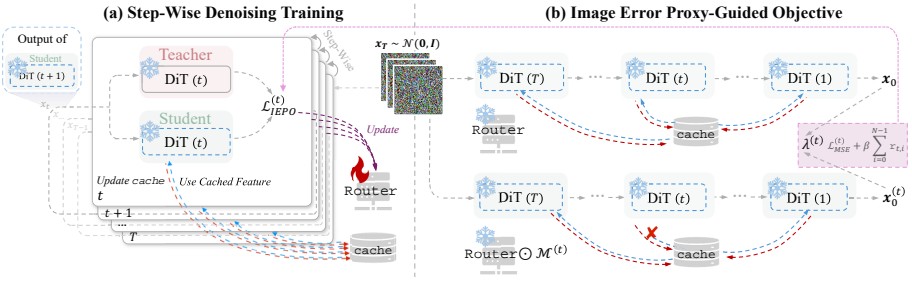

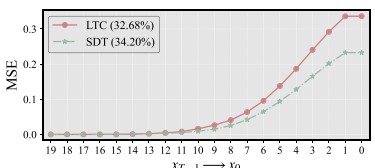

Figure 4. Overview of HarmoniCa. (a) *Step-Wise Denoising Training* (SDT) mimics the multi-timestep inference stage, which integrates the impact of prior timesteps at $t$. (b) *Image-Error Proxy-Guided Objective* (IEPO) incorporates the final image error into the learning objective by an efficient proxy $\lambda^{(t)}$, which is updated through *gradient-free* image generation passes every C training iterations. $\mathcal{M}^{(t)}$ masks the Router to disable the impact of the caching mechanism at $t$. $\odot$ denotes the element-wise multiplication. See detailed algorithms in Sec. A.

Figure 5. The Mean Square Error (MSE) of $x_t$ for DiT-XL/2 $256 \times 256$ (Peebles & Xie, 2023) induced by different feature caching methods. $x_t$ is the noisy image obtained at timestep $t+1$. "LTC" denotes Learning-to-Cache. For a fair comparison, $\mathcal{L}^{(t)}_{LTC}$ is employed for SDT. CUR is marked in the brackets.

## 4. HarmoniCa

In this section, we first observe that the existing learning-based caching shows discrepancies between the training and inference (Sec. 4.1). Then, we propose a framework named HarmoniCa to harmonize them for better feature caching (Sec. 4.2). Finally, it shows better efficiency and applicability than the previous training-based method (Sec. 4.3).

### 4.1. Discrepancy between Training and Inference

Most previous approaches (Selvaraju et al., 2024; Chen et al., 2024b) for DiT set the Router in a heuristic way. To be adaptive, Learning-to-Cache (Ma et al., 2024a) employs a learnable Router [3]. However, we have identified two discrepancies between its training and inference as follows.

**Prior timestep disregard.** As illustrated in Fig. 2, the inference process employing feature caching at timestep $t$ is subject to the prior timesteps. For example, at timestep $t = 1$, the input $x_1$ has the error induced by reusing the cached features $c_0$ and $c_1$ at preceding timestep $t = 2$. Furthermore, reusing and updating features at earlier timesteps also shape the contents of the current cache.

However, Learning-to-Cache is unaffected by prior denoising steps during training. Specifically, for each iteration, as depicted in Fig. 3, it first uniformly samples a timestep $t$ akin to DDPM (Ho et al., 2020). It then pre-fills an empty cache at $t$ and proceeds to train $\text{Router}_{t-1,:}$ at subsequent $t-1$. Without being influenced by the caching mechanism from $T$ to $t+1$, this pattern incurs significant error accumulation (Arora et al., 2022; Schmidt, 2019) as demonstrated by the trends of red polyline in Fig. 5.

**Objective mismatch.** Moreover, we also find that Learning-to-Cache (Ma et al., 2024a) solely aimed at aligning the predicted noise at each denoising step during training. It

leverages the following learning objective at timestep $t$:

$$\mathcal{L}^{(t)}_{LTC} = \mathcal{L}^{(t)}_{MSE} + \beta \sum_{i=0}^{N-1} r_{t,i}, \quad (4)$$

where $\beta$ is a coefficient for the regularization term of the $\text{Router}_{t:}$ and $\mathcal{L}^{(t)}_{MSE}$ represents MSE between predicted noise of DiT with and without feature caching at $t$.

In contrast, the target for inference is to obtain a high-quality image $x_0$. Thus, $\mathcal{L}^{(t)}_{LTC}$ induces a target mismatch due to bypassing direct image optimization. This potentially results in optimization shift (Rezatofighi et al., 2019) and severe object distortion, as shown by Fig. 6 (a) *vs.* (b).

### 4.2. Harmonizing Training and Inference

Existing studies (Ning et al., 2023; Li et al., 2024b; Ning et al., 2024) on diffusion models also validate that discrepancies between training and inference can lead to error accumulation and result in performance degradation. To solve the problem, we introduce HarmoniCa, a new learning-based caching framework that harmonizes training and inference through the following two techniques.

**Step-wise denoising training.** To mitigate the first discrepancy, as shown in Fig. 4 (a), we propose a new training paradigm named *Step-Wise Denoising Traning* (SDT). This strategy completes the entire denoising process over $T$ timesteps, thereby accounting for the cache usage and update from all prior timesteps. Specifically, at timestep $T$, we randomly sample a Gaussian noise $x_T$ and perform a single denoising step to pre-fill the cache. Over the following $T-1$ timesteps, the student model, which employs feature caching, step-wise removes noise to generate an image. Concurrently, the teacher model executes the same task without utilizing the cache. Requiring the student to mimic the output representation of its teacher, we compute the loss function and perform back-propagation to update $\text{Router}_{t:}$ at each timestep $t$. To ensure that each $r_{t,i}$ is dif-

---
[3] $r_{t,i}$ in the Router is a learnable parameter.

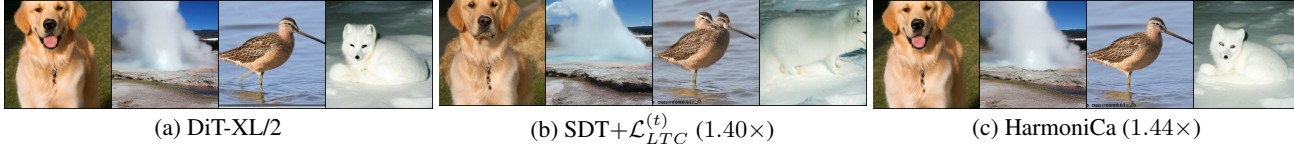

(a) DiT-XL/2        (b) SDT+$\mathcal{L}_{LTC}^{(t)}$ (1.40×)       (c) HarmoniCa (1.44×)

*Figure 6.* Random samples for DiT-XL/2 $256 \times 256$ *w/* and *w/o* feature caching ($T = 20$). We mark the speedup ratio in the brackets.

ferentiable during training (Ma et al., 2024a), distinct from Eq. (3), we proportionally combine the directly computed feature with the cached one to obtain $o_i$:

$$o_i = r_{t,i}b_i(\mathbf{h}_i, \mathbf{cs}) + (1 - r_{t,i})c_i. \quad (5)$$

Similar to inference, we also update $c_i$ in the `cache` with $b_i(\mathbf{h}_i, \mathbf{cs})$ when $r_{t,i} > \tau$. To improve training stability (Wimbauer et al., 2024), we fetch the output from the student as the input to the teacher for the next iteration. We repeat the above $T$ learning iterations during training.

As depicted in Fig. 5, by incorporating prior denoising timesteps during training, SDT significantly reduces accumulated error at each timestep and obtains a much more accurate $x_0$, with lower computation, compared to LTC.

**Image error proxy-guided objective.** For the second discrepancy, a straightforward solution to align the target with inference involves using the error of the final image $x_0$ caused by `cache` usage directly with a regularization term of `Router` as our training objective. However, even for DiT-XL/2 $256 \times 256$ (Peebles & Xie, 2023) with a small training batch size, this requires approximately $5\times$ GPU memory and $10\times$ time compared to SDT combined with $\mathcal{L}_{LTC}^{(t)}$ as detailed in Sec. B, making it impractical. Therefore, we have to identify a proxy for the error of the image $x_0$ that can be integrated into the learning objective.

Based on the above analysis, we propose an *Image Error Proxy-guided Objective* (IEPO). It is defined at each timestep $t$ as follows:

$$\mathcal{L}_{IEPO}^{(t)} = \lambda^{(t)}\mathcal{L}_{MSE}^{(t)} + \beta \sum_{i=0}^{N-1} r_{t,i}, \quad (6)$$

where $\lambda^{(t)}$ is our final image error proxy treated as a coefficient of $\mathcal{L}_{MSE}^{(t)}$. This proxy represents the final image error caused by the `cache` usage at $t$. With a large $\lambda^{(t)}$, $\mathcal{L}_{MSE}^{(t)}$ prioritizes reduction of the output error at $t$. This tends to decrease the cached feature usage rate at the corresponding timestep, and vice versa. Therefore, our proposed objective considers the trade-off between the error of $x_0$ and the `cache` usage at a certain denoising step.

Here, we detail the process to obtain $\lambda^{(t)}$. For a given `Router`, a mask matrix is defined to disable reusing cached features and force updating the entire `cache` at $t$ as:

$$\mathcal{M}_{j,k}^{(t)} = \begin{cases} 1, & j \neq t \\ \frac{1}{r_{j,k}}, & j = t \end{cases}, \quad (7)$$

where $(j, k)$ [4] denotes the index of $\mathcal{M}^{(t)} \in \mathbb{R}^{T \times N}$. As depicted in Fig. 4 (b), $x_0$ and $x_0^{(t)}$ are final images generated from a randomly sampled Gaussian noise $x_T$ using feature caching guided by (Upper) `Router` and (Lower) `Router` element-wise multiplied by $\mathcal{M}^{(t)}$, respectively. Then, we can formulate $\lambda^{(t)}$ as:

$$\lambda^{(t)} = \|x_0 - x_0^{(t)}\|_F^2, \quad (8)$$

where $\| \cdot \|_F$ denotes the Frobenius norm. To adapt to the training dynamics, we periodically update all the coefficients $\{\lambda^{(1)}, \ldots, \lambda^{(T)}\}$ every `C` iterations, where `C` mod $T = 0$, instead of employing static ones.

Fig. 6 shows that $\mathcal{L}_{IEPO}^{(t)}$ achieves significant performance improvement and yields accurate objective-level traits at a higher speedup ratio than $\mathcal{L}_{LTC}^{(t)}$. The study in Sec. C justifies that employing $\mathcal{L}_{LTC}^{(t)}$ incurs the optimization deviation from minimizing the error of $x_0$.

### 4.3. Efficiency Discussion

**Training efficiency.** HarmoniCa incurs significantly lower training costs than the previous learning-based method. In Tab. 1, HarmoniCa requires no training images (*i.e.*, *image-free*), whereas Learning-to-Cache utilizes original training datasets. Thus, it is challenging to apply Learning-to-Cache to models like the PIXART-$\alpha$ (Chen et al., 2023) family, which are trained on large datasets, limiting its applicability. Moreover, while dynamic update of $\lambda^{(t)}$ incurs approximately 10% extra time overhead, HarmoniCa requires only $\frac{3}{4}$ training hours compared to Learning-to-Cache, which needs to pre-fill the `cache` for each training iteration.

*Table 1.* Training costs of learning-based feature caching for DiT-XL/2 $256 \times 256$ ($T = 20$). We train with all methods for 20K iterations using a global batch size 64 on 4 NVIDIA H800 80GB GPUs. We set `C = 500`. Learning-to-Cache uses the full ImageNet training set (Russakovsky et al., 2015) as its original paper.

| Method | #Images | Time(h) | Memory(GB/GPU) |
|---|---|---|---|
| Learning-to-Cache | 1.22M | 2.15 | 33.33 |
| SDT+$\mathcal{L}_{LTC}^{(t)}$ | 0 | 1.47 | 33.28 |
| HarmoniCa | 0 | 1.63 | 33.28 |

**Inference efficiency.** For inference, our method with a pre-learned `Router` has little computational overhead during

---

[4] $1 \leq j \leq T$ and $0 \leq k \leq N - 1$.

runtime. Additionally, less than 6% extra memory overhead [5] is induced by `cache` for DiT-XL/2 $256 \times 256$. Therefore, the introduced cost is controlled at a small level. For PIXART-$\alpha$, HarmoniCa achieves over $2.07\times$ theoretical speedup [6] and $1.69\times$ real-world speedup with hugely improved performance than the non-accelerated one.

We provide more training and inference costs in Sec. D.

## 5. Experiments

### 5.1. Implementation Details

**Models and datasets.** We conduct experiments on two different image generation tasks. For class-conditional task, we employ DiT-XL/2 (Peebles & Xie, 2023) $256 \times 256$ and $512 \times 512$ pre-trained on ImageNet dataset (Russakovsky et al., 2015). For text-to-image (T2I) task, we utilize PIXART-$\alpha$ (Chen et al., 2023) series, known for its outstanding performance. These models, including PIXART-XL/2 at resolutions of $256 \times 256$ and $512 \times 512$, along with PIXART-XL/2-1024-MS at a higher resolution of $1024 \times 1024$, are tested on the MS-COCO dataset (Lin et al., 2015).

**Training settings.** Following (Ma et al., 2024a), we set the threshold $\tau$ as 0.1 for all the models. Each of them is trained for 20K iterations employing the AdamW optimizer (Loshchilov & Hutter, 2019) on 4 NVIDIA H800 80GB GPUs. The learning rate is fixed at 0.01, C is set to 500, and global batch sizes of 64, 48, and 32 are utilized for models with increasing resolutions. Additionally, we collect 1000 MS-COCO captions for T2I training.

**Baselines.** For class-conditional experiments, we choose the current SOTA Learning-to-Cache (Ma et al., 2024a) as our baseline. Due to the high training cost mentioned in Sec. 4.3, we employ FORA (Selvaraju et al., 2024) and $\Delta$-DiT (Chen et al., 2024b), excluding Learning-to-Cache for the T2I task. The results of these methods are obtained either by re-running their open-source code (if available) or by using the data provided in the original papers, all under the same conditions as our experiments. We also report the performance of models with reduced denoising steps. For a fair comparison, we exclude methods (Zou et al., 2024; Lou et al., 2024) employing highly different caching granularity, which is not the focus of the work, from our baselines.

**Evaluation.** To assess the generation quality, Fréchet Inception Distance (FID) (Nash et al., 2021), and sFID (Nash et al., 2021) are applied to all experiments. For DiT/XL-2, we additionally provide Inception Score (IS) (Salimans et al., 2016), Precision, and Recall (Kynkäänniemi et al.,

2019) as reference metrics. For PIXART-$\alpha$, to gauge the compatibility of image-caption pairs, we calculate CLIP score (Hessel et al., 2022) using ViT-B/32 (Dosovitskiy et al., 2020) as the backbone. To evaluate the inference efficiency, we measure CUR [7] and inference latency. In detail, we sample 50K images adopting DDIM (Song et al., 2020a) for DiT-XL/2, and 30K images utilizing IDDPM (Nichol & Dhariwal, 2021), DPM-Solver++ (Lu et al., 2022b), and SA-Solver (Xue et al., 2024) for PIXART-$\alpha$. All of them use classifier-free guidance (`cfg`) (Ho & Salimans, 2022).

More implementation details can be found in Sec. E, and the qualitative experiments are available in Sec. N. In addition, we apply the trained `Router` to a different sampler from training during inference in Sec. L. The robustness of our methods has also been validated in Sec. M.

### 5.2. Main Results

**Class-conditional generation.** We begin our evaluation for DiT-XL/2 on ImageNet and compare HarmoniCa with current SOTA Learning-to-Cache (Ma et al., 2024a) and the approach employing fewer timesteps. The results in Tab. 2 show that our method surpasses all baselines. Notably, with a higher speedup ratio for a 10-step DiT-XL/2 $256 \times 256$, HarmoniCa achieves an FID of 13.35 and an IS of 151.83, outperforming Learning-to-Cache by 1.24 and 6.74, respectively. Moreover, the superiority of our HarmoniCa increases as the number of timesteps decreases. We conjecture that it is because the difficulty to learn a `Router` rises as the timestep goes up. Additionally, we further conduct experiments with a lower CUR for this task in Sec. H.

*Table 2.* Accelerating generation on ImageNet for the DiT-XL/2. We highlight the best score in **bold**.

| Method | T | IS↑ | FID↓ | sFID↓ | Prec.↑ | Recall↑ | CUR(%)↑ | Latency(s)↓ |
|---|---|---|---|---|---|---|---|---|
| DiT-XL/2 $256 \times 256$ (`cfg` = 1.5) | | | | | | | | |
| DDIM (Song et al., 2020a) | 50 | 240.37 | 2.27 | 4.25 | 80.25 | 59.77 | - | 1.767 |
| DDIM (Song et al., 2020a) | 39 | 237.84 | 2.37 | 4.32 | 80.22 | 59.31 | - | $1.379_{(1.28\times)}$ |
| Learning-to-Cache (Ma et al., 2024a) | 50 | 233.26 | 2.62 | 4.50 | 79.40 | 59.15 | 23.39 | $1.419_{(1.25\times)}$ |
| HarmoniCa | 50 | **238.74** | **2.36** | **4.24** | **80.57** | **59.68** | 23.68 | $\mathbf{1.361}_{(1.30\times)}$ |
| DDIM (Song et al., 2020a) | 20 | 224.37 | 3.52 | 4.96 | 78.47 | 58.33 | - | 0.658 |
| DDIM (Song et al., 2020a) | 14 | 201.83 | 5.77 | 6.61 | 75.14 | 55.08 | - | $0.466_{(1.41\times)}$ |
| Learning-to-Cache (Ma et al., 2024a) | 20 | 201.37 | 5.34 | 6.36 | 75.04 | 56.09 | 35.60 | $0.468_{(1.41\times)}$ |
| HarmoniCa | 20 | **206.57** | **4.88** | **5.91** | **75.20** | **58.74** | 37.50 | $\mathbf{0.456}_{(1.44\times)}$ |
| DDIM (Song et al., 2020a) | 10 | 159.93 | 12.16 | 11.31 | 67.10 | 52.27 | - | 0.332 |
| DDIM (Song et al., 2020a) | 9 | 140.37 | 16.54 | 14.44 | 62.63 | 50.08 | - | $0.299_{(1.11\times)}$ |
| Learning-to-Cache (Ma et al., 2024a) | 10 | 145.09 | 14.59 | 11.58 | 64.03 | 52.06 | 19.11 | $0.279_{(1.19\times)}$ |
| HarmoniCa | 10 | **151.83** | **13.35** | **11.13** | **65.22** | **52.18** | 22.86 | $\mathbf{0.270}_{(1.23\times)}$ |
| DiT-XL/2 $512 \times 512$ (`cfg` = 1.5) | | | | | | | | |
| DDIM (Song et al., 2020a) | 20 | 184.47 | 5.10 | 5.79 | 81.77 | 54.50 | - | 3.356 |
| DDIM (Song et al., 2020a) | 16 | 173.31 | 6.47 | 6.67 | 81.10 | 51.30 | - | $2.688_{(1.25\times)}$ |
| Learning-to-Cache (Ma et al., 2024a) | 20 | 178.11 | 6.24 | 7.01 | 81.21 | 53.30 | 23.57 | $2.633_{(1.28\times)}$ |
| HarmoniCa | 20 | **179.84** | **5.72** | **6.61** | **81.33** | **55.80** | 25.98 | $\mathbf{2.574}_{(1.30\times)}$ |

**T2I generation.** We also present PixArt-$\alpha$ results in Tab. 3, comparing our HarmoniCa against FORA (Selvaraju et al.,

---

[5] The `cache` occupies 0.49 GB GPU memory with a batch size of 8 and original inference takes 8.18 GB GPU memory.

[6] The theoretical speedup is based on floating-point operations (FLOPs), and the real-world speedup is shown in Fig. 3.

[7] Definition can be found in Sec. 3.

*Table 3.* Accelerating generation on MS-COCO for PIXART-$\alpha$. The results of PIXART-$\Sigma$ (Chen et al., 2024a) family are available in Sec. F, including generation with resolution of $2048 \times 2048$.

| Method | T | CLIP↑ | FID↓ | sFID↓ | CUR(%)↑ | Latency(s)↓ |
|---|---|---|---|---|---|---|
| PIXART-$\alpha$ 256 × 256 (cfg = 4.5) | | | | | | |
| DPM-Solver++ (Lu et al., 2022b) | 20 | 30.96 | 27.68 | 36.39 | - | 0.553 |
| DPM-Solver++ (Lu et al., 2022b) | 15 | 30.77 | 31.68 | 38.92 | - | 0.418$_{(1.32\times)}$ |
| FORA (Selvaraju et al., 2024) | 20 | 31.10 | 27.42 | 37.98 | 50.00 | 0.364$_{(1.52\times)}$ |
| HarmoniCa | 20 | **31.13** | **26.33** | **37.85** | **56.01** | **0.346$_{(1.60\times)}$** |
| IDDPM (Nichol & Dhariwal, 2021) | 100 | 31.25 | 24.15 | 33.65 | - | 2.572 |
| IDDPM (Nichol & Dhariwal, 2021) | 75 | 31.25 | 24.17 | 33.73 | - | 1.868$_{(1.37\times)}$ |
| FORA (Selvaraju et al., 2024) | 100 | 31.25 | 25.16 | 33.62 | 50.00 | 1.558$_{(1.65\times)}$ |
| HarmoniCa | 100 | **31.17** | **23.73** | **32.23** | **53.24** | **1.523$_{(1.69\times)}$** |
| SA-Solver (Xue et al., 2024) | 25 | 31.31 | 26.78 | 38.35 | - | 0.891 |
| SA-Solver (Xue et al., 2024) | 20 | 31.23 | 27.45 | 39.01 | - | 0.665$_{(1.34\times)}$ |
| HarmoniCa | 25 | **31.27** | **27.07** | **38.62** | **54.19** | **0.561$_{(1.59\times)}$** |
| PIXART-$\alpha$ 512 × 512 (cfg = 4.5) | | | | | | |
| DPM-Solver++ (Lu et al., 2022b) | 20 | 31.30 | 23.96 | 40.34 | - | 1.759 |
| DPM-Solver++ (Lu et al., 2022b) | 15 | **31.29** | 25.12 | 40.37 | - | 1.291$_{(1.36\times)}$ |
| HarmoniCa | 20 | **31.29** | **24.81** | **40.18** | **54.64** | **1.072$_{(1.64\times)}$** |
| SA-Solver (Xue et al., 2024) | 25 | 31.23 | 25.43 | 39.84 | - | 2.263 |
| SA-Solver (Xue et al., 2024) | 20 | 31.19 | 25.85 | 40.08 | - | 1.738$_{(1.30\times)}$ |
| HarmoniCa | 25 | **31.20** | **25.74** | **39.99** | **54.24** | **1.406$_{(1.61\times)}$** |
| PIXART-$\alpha$ 1024 × 1024 (cfg = 4.5) | | | | | | |
| DPM-Solver++ (Lu et al., 2022b) | 20 | 31.10 | 25.01 | 37.80 | - | 9.470 |
| DPM-Solver++ (Lu et al., 2022b) | 15 | 31.07 | 25.77 | 42.50 | - | 7.141$_{(1.32\times)}$ |
| HarmoniCa | 20 | **31.09** | **23.02** | **36.24** | **55.06** | **5.786$_{(1.63\times)}$** |
| SA-Solver (Xue et al., 2024) | 25 | 31.05 | 23.65 | 38.12 | - | 11.931 |
| SA-Solver (Xue et al., 2024) | 20 | 31.02 | 23.88 | 39.41 | - | 9.209$_{(1.30\times)}$ |
| Harmonica | 25 | **31.07** | **23.77** | **38.93** | **53.98** | **7.551$_{(1.58\times)}$** |

*Table 4.* Evaluation with additional metrics for PIXART-$\alpha$.

| Method | T | DINO↑ | HPSv2↑ | PickScore↑ | CUR(%)↑ | Latency(s)↓ |
|---|---|---|---|---|---|---|
| PIXART-$\alpha$ 256 × 256 (cfg = 4.5) | | | | | | |
| DPM-Solver++ (Lu et al., 2022b) | 20 | 0.3082 | 28.91 | 27.89 | - | 0.553 |
| DPM-Solver++ (Lu et al., 2022b) | 15 | 0.2582 | 27.98 | 23.02 | - | 0.418$_{(1.32\times)}$ |
| FORA (Selvaraju et al., 2024) | 20 | 0.2712 | 28.11 | 22.44 | 50.00 | 0.364$_{(1.52\times)}$ |
| HarmoniCa | 20 | **0.3235** | **28.72** | **26.65** | **56.01** | **0.346$_{(1.60\times)}$** |
| PIXART-$\alpha$ 512 × 512 (cfg = 4.5) | | | | | | |
| DPM-Solver++ (Lu et al., 2022b) | 20 | 0.3339 | 30.53 | 28.52 | - | 1.759 |
| DPM-Solver++ (Lu et al., 2022b) | 15 | 0.3127 | 29.79 | 22.03 | - | 1.291$_{(1.36\times)}$ |
| FORA (Selvaraju et al., 2024) | 20 | 0.3099 | 29.82 | 21.98 | 50.0 | 1.150$_{(1.53\times)}$ |
| HarmoniCa | 20 | **0.3289** | **30.28** | **27.47** | **54.64** | **1.072$_{(1.64\times)}$** |

in LFM (Dao et al., 2023). As shown in Tab. 5 ($T = 100$, Euler Solver, and a batch size of 8), it achieves substantial speedup ratios without performance degradation.

*Table 5.* Accelerating generation for LFM (Dao et al., 2023).

| Dataset | Model | FID↑ | Latency(s)↓ |
|---|---|---|---|
| LSUN-Bedroom 256 × 256 (Yu et al., 2016) | DiT-L/2 | 5.22 | 1.76 |
| *w/* HarmoniCa | DiT-L/2 | 5.13 | 1.09$_{(1.62\times)}$ |
| CelebaHQ 256 × 256 (Karras et al., 2018) | DiT-L/2 | 5.42 | 1.76 |
| *w/* HarmoniCa | DiT-L/2 | 5.41 | 1.07$_{(1.65\times)}$ |
| ImageNet 256 × 256 (Russakovsky et al., 2015) | DiT-B/2 (cfg = 1.5) | 5.06 | 1.37 |
| *w/* HarmoniCa | DiT-B/2 (cfg = 1.5) | 5.04 | 0.87$_{(1.58\times)}$ |

**Comparison with the increase in speedup ratio.** To emphasize the significant advantage of our method over Learning-to-Cache, we present the IS and FID results as the speedup ratio increases for both Learning-to-Cache and our methods in Fig. 7. As the speedup ratio grows, the gap between Learning-to-Cache and our approach widens substantially. Specifically, with a speedup ratio of approximately 1.6, HarmoniCa achieves substantially higher IS and lower FID scores, 30.90 and 12.34, respectively, compared to Learning-to-Cache.

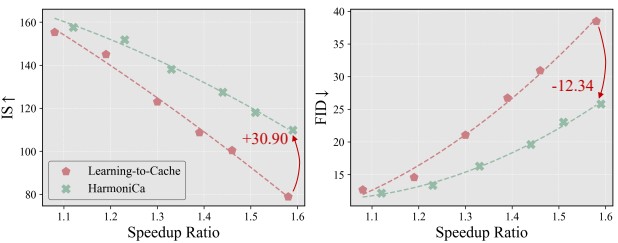

*Figure 7.* IS/FID with the increase of the speedup ratio for different methods. We employ DiT-XL/2 with a 10-step DDIM sampler on ImageNet 256 × 256.

**Comparison with additional feature caching methods.** To highlight HarmoniCa's advantages, we compare it with DeepCache (Ma et al., 2024b) and Faster Diffusion (Li et al., 2023a) on a single A6000 GPU. Due to the partial open-sourcing of the compared methods and the lack of implementation details, we directly report their results

2024) and the method using fewer timesteps. HarmoniCa outperforms these baselines across all metrics. For example, with the 20-step DPM-Solver++, PIXART-$\alpha$ $256 \times 256$ employing HarmoniCa achieves an FID of 26.33 and speeds up by $1.60\times$, surpassing the non-accelerated model's FID of 27.68. In contrast, DPM-Solver++ with 15 steps and FORA only achieve FIDs of 31.68 and 27.42, respectively, with speed increases under $1.52\times$. Notably, HarmoniCa also cuts about 41% off processing time without dropping performance when using the IDDPM sampler, while FORA results in over a 1.10 FID increase compared with the non-accelerated model. Overall, our method consistently delivers superior performance and speedup improvements across different resolutions and samplers, demonstrating its efficacy. Additionally, in Sec. I, HarmoniCa also significantly outperforms $\Delta$-DiT (Chen et al., 2024b).

Besides the above evaluation, we have also conducted experiments with more metrics (*e.g.*, Image-Reward (Xu et al., 2024), LPIPS (Zhang et al., 2018), and PSNR), which are provided in Sec. K. Here, we present DINO (Caron et al., 2021) and human evaluations (*e.g.*, HPSv2 (Wu et al., 2023) and PickScore (Kirstain et al., 2023)). As shown in the Tab. 4, our method outperforms baselines and achieves comparable performance with non-accelerated models.

**Results for flow-based models.** HarmoniCa can be effectively applied to rectified flow models (Liu et al., 2022b). We employ the pretrained models and evaluation

from Learning-to-Cache. As demonstrated in Tab. 6, HarmoniCa shows a negligible < 0.05 FID increase with a 1.65× speedup ratio, outperforming both methods. Notably, DeepCache is constrained by the *U-shaped structure*, making it unsuitable for DiTs.

*Table 6.* Comparison between different caching-based approaches. We use U-ViT (Bao et al., 2023) on ImageNet 256×256 here.

| Method | T | FID↓ | Latency(s)↓ |
|---|---|---|---|
| DPM-Solver (Lu et al., 2022a) | 20 | 2.57 | 7.60 |
| Faster Diffusion (Li et al., 2023a) | 20 | 2.82 | 5.95(1.28×) |
| DeepCache (Ma et al., 2024b) | 20 | 2.70 | 4.68(1.62×) |
| HarmoniCa | 20 | **2.61** | **4.60(1.65×)** |

*Table 7.* Comparison between different acceleration approaches. We use DiT-XL/2 on ImageNet 256×256 here. "*" denotes the latency was tested on one NVIDIA A100 80GB GPU. Experimental details are presented in Sec. G.

| Method | T | IS↑ | FID↓ | sFID↓ | Latency(s)↓ | Latency(s)↓* |
|---|---|---|---|---|---|---|
| DDIM (Zhang et al., 2022) | 20 | 224.37 | 3.52 | 4.96 | 0.658 | 1.217 |
| EfficientDM (He et al., 2024) | 20 | 172.70 | 6.10 | **4.55** | 0.591(1.11×) | 0.842(1.45×) |
| PTQ4DiT (Wu et al., 2024) | 20 | 17.06 | 71.82 | 23.16 | 0.577(1.14×) | 0.839(1.45×) |
| Diff-Pruning (Fang et al., 2023) | 20 | 168.10 | 8.22 | 6.20 | 0.458(1.44×) | **0.813(1.50×)** |
| HarmoniCa | 20 | **206.57** | **4.88** | 5.91 | **0.456(1.44×)** | 0.815(1.49×) |

**Comparison with pruning and quantization.** As shown in Tab. 7, we compare our HarmoniCa with advanced quantization and pruning methods. Our method significantly outperforms these methods, demonstrating the substantial benefit of feature caching for accelerating DiT models. It is important to note that the speedup ratio for quantization is partially determined by hardware support, which we do not rely on, and the current customized CUDA kernel often lacks optimization on H800's *Hopper architecture*. Here, we believe the significant performance drop of PTQ4DiT results from small sampling steps. A 50/250-step DDPM sampler is used in the original paper.

**Combination with quantization.** We conduct experiments to show the high compatibility of our HarmoniCa with the model quantization technique. In Tab. 8, our method boosts a considerable speedup ratio from 1.18× to 1.85× with only a 0.12 FID increase for PIXART-$\alpha$ 256 × 256. In the future, we will explore combining our HarmoniCa with other acceleration techniques, such as pruning and distillation, to further reduce the computational demands for DiT.

## 5.3. Ablation Studies

In this subsection, we employ a 20-step DDIM (Song et al., 2020a) sampler for DiT-XL/2 256 × 256.

**Effect of different components.** To show the effectiveness of components involved in HarmoniCa, we apply different combinations of training techniques and show the results in Tab. 9. For the training paradigm, equipped with $\mathcal{L}_{LTC}^{(t)}$,

*Table 8.* Results of combining our framework with an advanced quantization method: EfficientDM (He et al., 2024). IS↑ is for DiT-XL/2 and CLIP↑ is for PIXART-$\alpha$ in the table. Experimental details here can be found in Sec. G.

| Method | IS↑/CLIP↑ | FID↓ | sFID↓ | CUR(%)↑ | Latency(s)↓ | #Size(GB)↓ |
|---|---|---|---|---|---|---|
| DiT-XL/2 256 × 256 (cfg = 1.5) | | | | | | |
| EfficientDM (He et al., 2024) | 172.70 | 6.10 | 4.55 | - | 0.591(1.11×) | 0.64(3.93×) |
| w/ HarmoniCa ($\beta = 4e^{-8}$) | 168.16 | 6.48 | 4.32 | 26.25 | 0.473(1.40×) | 0.64(3.93×) |
| PIXART-$\alpha$ 256 × 256 (cfg = 4.5) | | | | | | |
| EfficientDM (He et al., 2024) | 30.09 | 34.84 | 30.34 | - | 0.469(1.18×) | 0.59(1.98×) |
| w/ HarmoniCa | 30.15 | 34.96 | 30.55 | 53.34 | 0.299(1.85×) | 0.59(1.98×) |
| PIXART-$\alpha$ 512 × 512 (cfg = 4.5) | | | | | | |
| EfficientDM (He et al., 2024) | 30.71 | 25.82 | 41.64 | - | 0.461(1.20×) | 0.59(1.98×) |
| w/ HarmoniCa | 30.75 | 26.15 | 41.99 | 53.11 | 0.281(1.97×) | 0.59(1.98×) |

our SDT significantly decreases FID by 10 compared to that of Learning-to-Cache. For the learning objective, our IEPO achieves nearly a 40 IS improvement and a 3.13 FID reduction for SDT compared with $\mathcal{L}_{LTC}^{(t)}$. Moreover, both SDT and IEPO can help significantly enhance performance for the counterparts in the table. For a fair comparison, we modify the implementation of Learning-to-Cache to train the entire Router in Tab. 9. A detailed discussion of this can be found in Sec. J.

*Table 9.* Effect of different components. The first row denotes the model *w/o* feature caching. The green and blue rows denote Learning-to-Cache and HarmoniCa, respectively.

| Training Paradigm | | Learning Objective | | IS↑ | FID↓ | sFID↓ | CUR(%)↑ | Latency(s)↓ |
|---|---|---|---|---|---|---|---|---|
| Learning-to-Cache | SDT | $\mathcal{L}_{LTC}^{(t)}$ | $\mathcal{L}_{IEPO}^{(t)}$ | | | | | |
| | | | | 224.37 | 3.52 | 4.96 | - | 0.658 |
| ✔ | | ✔ | | 115.00 | 18.57 | 16.18 | 32.68 | 0.483(1.36×) |
| ✔ | | | ✔ | 203.41 | 5.20 | 6.07 | 36.70 | 0.458(1.44×) |
| | ✔ | ✔ | | 166.65 | 8.01 | 7.62 | 34.20 | 0.471(1.40×) |
| | ✔ | | ✔ | **206.67** | **4.88** | **5.91** | **37.50** | **0.456(1.44×)** |

**Effect of iteration interval** C**.** As illustrated in Fig. 8, we carry out experiments to evaluate the impact of different values of C on updating $\lambda^{(t)}$ in Eq. (8). Despite the similar

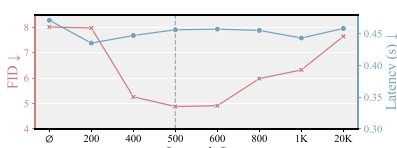

*Figure 8.* Ablation results of different iteration interval C. "∅" denotes the employing $\mathcal{L}_{LTC}^{(t)}$ as loss function.

speedup ratios, employing an extreme C value leads to notable performance drops. Specifically, a large C means the proxy $\lambda^{(t)}$ fails to accurately and timely reflect the caching mechanism's effect on the final image. Conversely, a small C results in overly frequent updates, complicating training convergence. Hence, we choose a moderate value of 500 as C in this paper based on its superior performance.

*Table 10.* Ablation results of different metrics for $\lambda^{(t)}$. The first and second columns represent the model *w/o* feature caching and SDT+$\mathcal{L}_{LTC}^{(t)}$, respectively. $\mathcal{D}_{KL}(\cdot)$ denotes Kullback–Leibler (KL) divergence. We employ $\|\cdot\|_F^2$ in the paper to obtain $\lambda^{(t)}$.

| $\lambda^{(t)}$ | $+\infty$ | 1 | $\sum\|\boldsymbol{x}_0 - \boldsymbol{x}_0^{(t)}\|$ | $\|\boldsymbol{x}_0 - \boldsymbol{x}_0^{(t)}\|_F^2$ | $\mathcal{D}_{KL}(\boldsymbol{x}_0, \boldsymbol{x}_0^{(t)})$ | $1 - \text{MS-SSIM}(\boldsymbol{x}_0, \boldsymbol{x}_0^{(t)})$ | $\text{LPIPS}(\boldsymbol{x}_0, \boldsymbol{x}_0^{(t)})$ |
|---|---|---|---|---|---|---|---|
| IS↑ | 224.37 | 166.65 | 172.08 | **206.57** | 205.91 | 204.72 | 205.83 |
| FID↓ | 3.52 | 8.01 | 6.95 | 4.88 | 5.25 | 4.91 | **4.83** |
| sFID↓ | 4.96 | 7.62 | 7.79 | 5.91 | **5.51** | 5.83 | 5.57 |
| CUR(%)↑ | - | 34.20 | 34.82 | **37.50** | 36.79 | 37.68 | 37.32 |
| Latency(s)↓ | 0.658 | 0.471$_{(1.40\times)}$ | 0.470$_{(1.40\times)}$ | **0.456**$_{(1.44\times)}$ | 0.458$_{(1.44\times)}$ | **0.456**$_{(1.44\times)}$ | **0.456**$_{(1.44\times)}$ |

*Table 11.* Performance of HarmoniCa across different values of $\tau \in [0, 1)$. $\tau$ is the threshold for the `Router` as described in Sec. 3.

| $\tau$ | T | IS↑ | FID↓ | sFID↓ | Latency(s)↓ |
|---|---|---|---|---|---|
| 0.1 | 10 | 151.83 | 13.35 | 11.13 | 0.270$_{(1.23\times)}$ |
| 0.5 | 10 | 151.80 | 13.41 | 11.09 | 0.269$_{(1.23\times)}$ |
| 0.9 | 10 | 151.78 | 13.37 | 11.08 | 0.270$_{(1.23\times)}$ |

**Effect of coefficient $\beta$.** We also explore the trade-off between inference speed and performance for various values of $\beta$ in Eq. (6). As shown in Fig. 9, a higher $\beta$ leads to

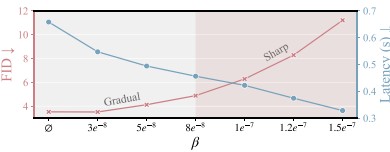

*Figure 9.* Ablation results of different coefficient $\beta$. "$\varnothing$" denotes the model *w/o* feature caching.

greater acceleration but at the cost of more pronounced performance degradation, and vice versa. Notably, performance declines gradually when $\beta \leq 8e^{-8}$ and more sharply outside this range. This observation suggests the potential for autonomously finding an optimal $\beta$ to balance speed and performance, which we aim to address in future research.

**Effect of different metrics for $\lambda^{(t)}$.** In Tab. 10, we conduct experiments to explore the effect of $\lambda^{(t)}$ with different metrics. Both $\|\cdot\|_F^2$ and $\mathcal{D}_{KL}(\cdot)$ lead to notable performance enhancements compared to using only the output error (*i.e.*, $\lambda^{(t)} = 1$) at each timestep. Due to the insensitivity to outliers, $\sum|\cdot|$ is generally less effective for image reconstruction and inferior to the others in the table. We further test MS-SSIM (Wang et al., 2003) and LPIPS [8] (Zhang et al., 2018), which are designed to evaluate natural image quality as metrics for $\lambda^{(t)}$. These metrics exhibit similar performance compared with $\|\cdot\|_F^2$.

**Effect of different threshold $\tau$.** we conduct study on different values of caching threshold $\tau$ in Tab. 11. The results demonstrate our method is robust *w.r.t* variations in $\tau$. Thus, we set $\tau$ to 0.1 for all the experiments in this work.

**SDT *vs.* teacher forcing.** Teacher forcing, which uses the outputs of a non-accelerated teacher model as the input data (*i.e.*, replace $\boldsymbol{\epsilon}^{(t)'}$ by $\boldsymbol{\epsilon}^{(t)}$ in the 13th row of Alg. 1) for the next iteration of training, may further help mitigate cumulative errors. In Tab. 12, SDT shows comparable results with teacher forcing, which indicates that using $\boldsymbol{\epsilon}^{(t)'}$ would not lead to potential error accumulation compared with $\boldsymbol{\epsilon}^{(t)}$. A more detailed theoretical analysis is planned for future work.

---

[8]AlexNet (Krizhevsky et al., 2017) is used to extract features.

*Table 12.* Comparison between SDT and teacher forcing. Both employ $\mathcal{L}_{IEPO}^{(t)}$ as their loss functions.

| Method | T | IS↑ | FID↓ | sFID↓ | Latency(s)↓ |
|---|---|---|---|---|---|
| DiT-XL/2 256 × 256 (`cfg = 1.5`) | | | | | |
| SDT | 20 | 4.88 | 206.57 | **5.91** | 0.456$_{(1.44\times)}$ |
| Teacher forcing | 20 | **4.87** | 207.12 | 6.02 | **0.458**$_{(1.44\times)}$ |
| DiT-XL/2 512 × 512 (`cfg = 1.5`) | | | | | |
| SDT | 20 | 5.72 | 179.84 | 6.61 | **2.574**$_{(1.30\times)}$ |
| Teacher forcing | 20 | 5.74 | 178.96 | **6.61** | 2.577$_{(1.30\times)}$ |

## 6. Conclusions and Limitations

In this research, we focus on accelerating Diffusion Transformers (DiTs) through a learning-based caching mechanism. We first identify two discrepancies between training and inference of the previous method: (1) *Prior Timestep Disregard* in which earlier step influences are neglected during training, and (2) *Objective Mismatch*, where training only focuses on intermediate results. To solve these, we introduce a novel caching framework named **HarmoniCa**, which consists of *Step-wise Denoising Training* (SDT) and an *Image Error Proxy-Guided Objective* (IEPO). SDT captures the influence of all timesteps during training, while IEPO introduces an efficient proxy for image error. Extensive experiments show that HarmoniCa achieves superior performance and efficiency with lower training costs than the existing training-based method. In terms of limitations, we focus on *block-wise* caching and image generation in this work. However, we believe our work is easy to expand to any caching granularity and video/audio/3D generation.

## Acknowledgement

We thank Chengtao Lv and Yuyang Chen for their insights and feedback, and Yifu Ding for her help with diagrams. This work was supported by the Hong Kong Research Grants Council under the Areas of Excellence scheme grant AoE/E-601/22-R and NSFC/RGC Collaborative Research Scheme grant CRS_HKUST603/22. It was also supported by the Beijing Municipal Science and Technology Project (No. Z231100010323002), the National Natural Science Foundation of China (Nos. 62306025, 92367204), CCF-Baidu Open Fund, and Beijing Natural Science Foundation (QY24138).

## Impact Statement

This paper presents work whose goal is to advance the field of Machine Learning. There are many potential societal consequences of our work, none which we feel must be specifically highlighted here.

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

## A. Alogrithm of HarmoniCa

As described in Alg. 1, we provide a detailed algorithm of our HarmoniCa. For clarity, we omit the pre-fill stage (*i.e.*, denoising at $T$), where $\texttt{Router}_{T:}$ is forced to be set to $\{1\}_{1 \times N}$. The $\texttt{conds}$ for T2I tasks and class-conditional generation are pre-prepared text prompts and class labels, respectively.

---

**Algorithm 1** HarmoniCa: the upper snippet describes the full procedure, and the lower side contains the subroutine for computing the proxy of the final image error.

---

func HARMONICA($\phi, \boldsymbol{\epsilon}_\theta, \texttt{iters}, \texttt{conds}, \tau, \beta, T, \texttt{C}$)

**Require:** $\phi(\cdot)$ — diffusion sampler
         $\boldsymbol{\epsilon}_\theta(\cdot)$ — DiT model
         $\texttt{iters}$ — amount of training iterations
         $\texttt{conds}$ — conditional inputs
         $\tau$ — threshold
         $\beta$ — constraint coefficient
         $T$ — maximum denoising step
         $\texttt{C}$ — iteration interval

1: Initialize $\texttt{Router}$ with a normal distribution
2: $\texttt{cache} = \emptyset$          ▷ Initialize $\texttt{cache}$
3: **for** $i$ in 0 to $\frac{\texttt{iters}}{T} - 1$ **do**:
4:     $\boldsymbol{x}_T \sim \mathcal{N}(\mathbf{0}, \mathbf{I})$
5:     **if** $i \% \frac{\texttt{C}}{T} = 0$ **then**
6:        $\{\lambda^{(1)}, \ldots, \lambda^{(T)}\} = \texttt{gen\_proxy}(\phi, \boldsymbol{\epsilon}_\theta, \boldsymbol{x}_T, \texttt{conds}[i], \tau, \texttt{Router})$
7:     **end if**
8:     **for** $t$ in $T$ to 1 **do**:
9:        $\boldsymbol{\epsilon}^{(t)'} = \boldsymbol{\epsilon}_\theta(\boldsymbol{x}_t, t, \texttt{conds}[i], \texttt{Router}_{t,:}, \tau, \texttt{cache})$          ▷ Fig. 2
10:       $\boldsymbol{\epsilon}^{(t)} = \boldsymbol{\epsilon}_\theta(\boldsymbol{x}_t, t, \texttt{conds}[i])$
11:       $\mathcal{L}_{IEPO}^{(t)} = \lambda^{(t)} \|\boldsymbol{\epsilon}^{(t)'} - \boldsymbol{\epsilon}^{(t)}\|_F^2 + \beta \sum_{i=0}^{N-1} \text{r}_i^{(t)}$          ▷ Eq. (6)
12:       Tune $\texttt{Router}_{t,:}$ by back-propagation
13:       $\boldsymbol{x}_{t-1} = \phi(\boldsymbol{x}_t, t, \boldsymbol{\epsilon}^{(t)'})$
14:     **end for**
15: **end for**
16: **return** $\texttt{Router}$

func gen\_proxy($\phi, \boldsymbol{\epsilon}_\theta, \boldsymbol{x}_T, \texttt{cond}, \tau, \texttt{Router}$)          ▷ Wrapped by *stopgradient* operator
1: $\texttt{cache} = \emptyset$          ▷ Initialize $\texttt{cache}$
2: Employ feature cache guided by $\texttt{Router}$ to generate $\boldsymbol{x}_0$
3: **for** $t$ in $T$ to 1 **do**:
4:     Generate $\mathcal{M}^{(t)}$          ▷ Eq. (7)
5:     Employ feature cache guided by $\texttt{Router} \odot \mathcal{M}^{(t)}$ to generate $\boldsymbol{x}_0^{(t)}$
6:     $\lambda^{(t)} = \|\boldsymbol{x}_0 - \boldsymbol{x}_0^{(t)}\|_F^2$          ▷ Eq. (8)
7: **end for**
8: **return** $\{\lambda^{(1)}, \lambda^{(2)}, \ldots, \lambda^{(T)}\}$

---

## B. Image Error with **Router** Regularization Term as Training Objective

In Tab. 13, SDT+$\mathcal{L}_{\boldsymbol{x}_0}^{(t)}$ requires $t - 1$ additional denoising passes per training iteration at $t$ to compute the error of $\boldsymbol{x}_0$. Consequently, this approach consumes about $9.73\times$ GPU hours compared to SDT+$\mathcal{L}_{LTC}^{(t)}$. Due to the extensive intermediate activations stored from timestep $t$ to 1 for back-propagation, it also costs $4.90\times$ GPU memory. This estimation is conducted with small batch sizes and limited iterations. Therefore, SDT+$\mathcal{L}_{\boldsymbol{x}_0}^{(t)}$ is less feasible for models with larger latent spaces or higher token counts per image, such as DiT-XL/2 $512 \times 512$, particularly in large-batch, complete training scenarios. Additionally, the network effectively becomes $T \times N$ stacked Transformer blocks under this strategy, making it difficult (Wang et al., 2024a) to optimize

*Table 13.* Training costs estimation across different methods for DiT-XL/2 $256 \times 256$ (Peebles & Xie, 2023) ($T = 20$). We only employ 5K iterations with a global batch size of 8 on 4 NVIDIA H800 80G GPUs. $\mathcal{L}_{\boldsymbol{x}_0}^{(t)}$ denotes the loss function replacing $\mathcal{L}_{MSE}^{(t)}$ in Eq. (4) with the final image error.

| Method | #Images | Time(h) | Memory(GB/GPU) |
|---|---|---|---|
| SDT+$\mathcal{L}_{\boldsymbol{x}_0}^{(t)}$ | 0 | 1.46 | 65.36 |
| SDT+$\mathcal{L}_{LTC}^{(t)}$ | 0 | 0.15 | 13.33 |

the `Router` with even a moderate $T$ value, such as 50 or 100.

## C. Optimization Deviation

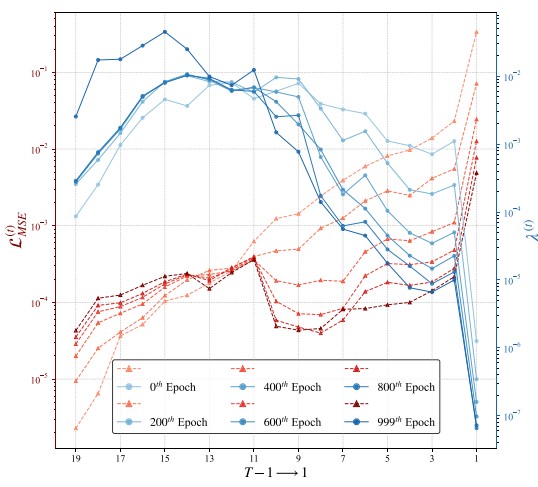

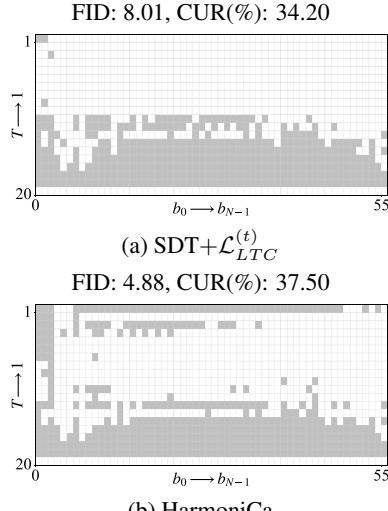

FID: 8.01, CUR(%): 34.20

(a) SDT+$\mathcal{L}_{LTC}^{(t)}$

FID: 4.88, CUR(%): 37.50

(b) HarmoniCa

*Figure 10.* (Left) Variations of $\mathcal{L}_{MSE}^{(t)}$ and $\lambda^{(t)}$ for SDT+$\mathcal{L}_{LTC}^{(t)}$. (Right) `Router` visualization across different methods. The gray grid $(t, i)$ represents using the feature in `cache` at $t$ without computing $o_i$. The white grid indicates computing and updating `cache`. We also mark their FID (Heusel et al., 2018) and CUR. All the above experiments employ DiT-XL/2 $256 \times 256$ ($T = 20, N = 56$).

To generate high-quality $x_0$ and accelerate the inference phase, we believe only considering the output error at a certain timestep can cause a deviated optimization due to its gap *w.r.t* the error of $x_0$. To validate this, we plot the values of $\mathcal{L}_{MSE}^{(t)}$ in Eq. (4) and $\lambda^{(t)}$ in Eq. (8) during the training phase of SDT+$\mathcal{L}_{LTC}^{(t)}$ in Fig. 10 (Left). Comparing $\mathcal{L}_{MSE}^{(t)}$ and $\lambda^{(t)}$ across different denoising steps, their results present a significant discrepancy. For instance, $\mathcal{L}_{MSE}^{(t)}$ at $t = 14$ is several orders of magnitude smaller than that at $t = 1$ during the entire training process, and the opposite situation happens for $\lambda^{(t)}$. Intuitively, this indicates that we could increase the cache usage rate at $t = 1$, and vice versa at $t = 14$ for higher performance while keeping the same speedup ratio according to the value of the proxy $\lambda^{(t)}$. However, only considering the output error at each timestep (*i.e.*, $\mathcal{L}_{MSE}^{(t)}$) can optimize towards a shifted direction. In practice, the learned `Router` with the guidance of $\lambda^{(t)}$ in Fig. 10 (Right) (b) caches less in large timesteps like $t = 14$ and reuses more in small timesteps as $t = 1$ compared to that in Fig. 10 (Right) (a) achieving significant performance enhancement.

## D. Training and Inference Costs for PIXART-$\alpha$

We provide the computation costs of training and inference for PIXART-$\alpha$ (Chen et al., 2023) in Tab. 14. It is worth noting that we only use naive distributed data parallel (DDP) training without any advanced strategies like gradient checkpointing (Chen et al., 2016) and FSDP (Zhao et al., 2023). Thus, the requirements of computation can be further decreased in practice.

*Table 14.* Training and inference costs.

| Method | Infer. Mem. (GB/GPU)↓ | Train. Mem. (GB/GPU)↓ | Train. Time (h)↓ |
|---|---|---|---|
| PIXART-$\alpha$ $256 \times 256$ (`cfg` = 4.5) | | | |
| DPM-Solver++ (Lu et al., 2022b) | 20.86 | - | - |
| HarmoniCa | 21.21 | 79.05 | 1.84 |
| PIXART-$\alpha$ $512 \times 512$ (`cfg` = 4.5) | | | |
| DPM-Solver++ (Lu et al., 2022b) | 24.11 | - | - |
| HarmoniCa | 24.61 | 77.30 | 2.92 |

## E. More Implementation Details

In this section, we present more details on the implementation of our HarmoniCa. First, following (Ma et al., 2024a), we also perform a sigmoid function [9] to each $r_{t,i}$ before it is passed to the model. Moreover, unless specified otherwise, the hyper-parameter $\beta$ in Eq. (6) for all experiments is given in Tab. 15; any exceptions are noted in the relevant tables.

*Table 15.* Hyper-parameter $\beta$ for training the `Router`.

| Model | DiT-XL/2 | | | | PIXART-$\alpha$ | | | | | PIXART-$\Sigma$ | | |
|---|---|---|---|---|---|---|---|---|---|---|---|---|
| Resolution | $256 \times 256$ | | | $512 \times 512$ | $256 \times 256$ | | | $512 \times 512$ | $1024 \times 1024$ | $512 \times 512$ | $1024 \times 1024$ | $2048 \times 2048$ |
| $T$ | 10 | 20 | 50 | 20 | 20 | 100 | 25 | 20 | 20 | 20 | 20 | 20 |
| $\beta$ | $7e^{-8}$ | $8e^{-8}$ | $5e^{-8}$ | $4e^{-8}$ | $1e^{-3}$ | $8e^{-4}$ | $8e^{-4}$ | $8e^{-4}$ | $8e^{-4}$ | $1e^{-3}$ | $8e^{-4}$ | $8e^{-4}$ |

## F. Results for PIXART-$\Sigma$

In this section, we present the results for the PIXART-$\Sigma$ family, including PIXART-$\Sigma$-XL/2-1024-MS and PIXART-$\Sigma$-XL/2-2K-MS. For the latter one, we test by sampling 10K images. Additionally, we train the `Router` with a batch size of 16 and measure latency using a batch size of 1. All other settings are consistent with those described in Sec. 5.1.

As shown in Table 16, HarmoniCa achieves a $1.56\times$ speedup along with improved CLP scores compared to the non-accelerated model for PIXART-$\Sigma$ $2048 \times 2048$. Notably, this is the *first time* for the feature caching mechanism to accelerate image generation with such a super-high resolution of $2048 \times 2048$.

*Table 16.* Accelerating image generation on MS-COCO for the PIXART-$\Sigma$.

| Method | T | CLIP↑ | FID↓ | sFID↓ | CUR(%)↑ | Latency(s)↓ |
|---|---|---|---|---|---|---|
| PIXART-$\Sigma$ $512 \times 512$ (`cfg` = 4.5) | | | | | | |
| DPM-Solver++ (Lu et al., 2022b) | 20 | 31.20 | 26.81 | 42.79 | - | 1.912 |
| DPM-Solver++ (Lu et al., 2022b) | 15 | 31.23 | 25.99 | 42.08 | - | $1.435_{(1.34\times)}$ |
| HarmoniCa | 20 | **31.28** | **24.64** | **41.58** | 53.45 | $\textbf{1.145}_{(1.67\times)}$ |
| PIXART-$\Sigma$ $1024 \times 1024$ (`cfg` = 4.5) | | | | | | |
| DPM-Solver++ (Lu et al., 2022b) | 20 | 31.37 | 20.98 | 27.47 | - | 9.467 |
| DPM-Solver++ (Lu et al., 2022b) | 15 | 31.34 | 21.63 | 28.68 | - | $7.100_{(1.33\times)}$ |
| HarmoniCa | 20 | **31.50** | **20.53** | **27.05** | 52.74 | $\textbf{5.852}_{(1.62\times)}$ |
| PIXART-$\Sigma$ $2048 \times 2048$ (`cfg` = 4.5) | | | | | | |
| DPM-Solver++ (Lu et al., 2022b) | 20 | 31.19 | 23.61 | 51.12 | - | 14.198 |
| DPM-Solver++ (Lu et al., 2022b) | 15 | 31.26 | 24.40 | 53.34 | - | $9.782_{(1.45\times)}$ |
| HarmoniCa | 20 | **31.51** | 24.09 | **51.83** | 53.80 | $\textbf{9.081}_{(1.56\times)}$ |

## G. Experimental Details for Quantization and Pruning

For EfficientDM (He et al., 2024), we employ 8-bit channel-wise weight quantization and 8-bit layer-wise activation quantization (Gong et al., 2024; 2019) for full-precision (FP32) DiT-XL/2 and half-precision (FP16) PIXART-$\alpha$. The former uses a 20-step DDIM sampler (Song et al., 2020a), while the latter employs a DPM-Solver++ sampler (Lu et al., 2022b) with the same steps. More specifically, we use MSE initialization (Nagel et al., 2021) for quantization parameters. For the quantization-aware fine-tuning stage, we set the learning rate of LoRA (Hu et al., 2021) and activation quantization parameters to $1e^{-6}$ and that of weight quantization parameters to $1e^{-5}$, respectively. Additionally, we employ 3.2K iterations for DiT-XL/2 (Peebles & Xie, 2023) and 9.6K iterations for PIXART-$\alpha$ (Chen et al., 2023) on a single NVIDIA H800 80G GPU. Other settings are the same as those from the original paper (He et al., 2024). Leveraging NVIDIA CUTLASS (Kerr et al., 2017) implementation, we evaluate the latency of quantized models employing the 8-bit multiplication for all the linear layers and convolutions. For PTQ4DiT, we implemented the DDIM sampler and re-run the open-source code, which originally only supported DDPM. For Diff-Pruning, we re-implement the method for the DiT model (which originally only supported U-Net models) and follow the settings specified in the original paper.

---

[9] $\sigma(x) = \frac{1}{1+e^{-x}}$

## H. Comparison between Learning-to-Cache and HarmoniCa with a Low CUR(%)

In this section, we compare HarmoniCa with Learning-to-Cache (Ma et al., 2024a) at a relatively low CUR(%). As shown in Tab. 17, both methods achieve a similar speedup ratio and even better performance than non-accelerated models. Therefore, we employ higher CUR in Tab. 2 to show our pronounced superiority.

*Table 17.* Comparison results between Learning-to-Cache and HarmoniCa for the DiT-XL/2 with a low CUR(%).

| Method | T | IS↑ | FID↓ | sFID↓ | Prec.↑ | Recall↑ | CUR(%)↑ | Latency(s)↓ |
|---|---|---|---|---|---|---|---|---|
| DiT-XL/2 256 × 256 (cfg = 1.5) | | | | | | | | |
| DDIM (Song et al., 2020a) | 20 | 224.37 | 3.52 | 4.96 | 78.47 | 58.33 | - | 0.658 |
| DDIM (Song et al., 2020a) | 15 | 214.77 | 4.17 | 5.54 | 77.43 | 56.30 | - | $0.564_{(1.17\times)}$ |
| Learning-to-Cache (Ma et al., 2024a) | 20 | 228.19 | **3.49** | **4.66** | 79.32 | 59.10 | **22.05** | $0.545_{(1.21\times)}$ |
| HarmoniCa ($\beta = 3e^{-8}$) | 20 | **228.79** | 3.51 | 4.76 | **79.43** | **59.32** | 21.07 | $0.547_{(1.20\times)}$ |
| DiT-XL/2 512 × 512 (cfg = 1.5) | | | | | | | | |
| DDIM (Song et al., 2020a) | 20 | 184.47 | 5.10 | 5.79 | 81.77 | 54.50 | - | 3.356 |
| DDIM (Song et al., 2020a) | 18 | 180.06 | 5.62 | 6.13 | 81.37 | 53.90 | - | $3.021_{(1.11\times)}$ |
| Learning-to-Cache (Ma et al., 2024a) | 20 | 183.57 | 5.45 | 6.05 | **82.10** | 54.90 | 14.64 | $2.927_{(1.15\times)}$ |
| HarmoniCa ($\beta = 2e^{-8}$) | 20 | **183.71** | **5.32** | **5.84** | 81.83 | **55.80** | 16.61 | $\mathbf{2.863}_{(1.17\times)}$ |

## I. Comparison between ∆-DiT and HarmoniCa

In this section, we compare HarmoniCa with ∆-DiT (Chen et al., 2024b). Given that the code and implementation details of ∆-DiT [10] are not open source, we report results derived from the original paper. Additionally, we evaluate performance sampling 5000 images as used in that study. As depicted in Tab 18, our framework further decreases 9.3% latency and gains 3.35 FID improvement compared with ∆-DiT for PIXART-$\alpha$ with a 20-step DPM-Solver++ sampler (Lu et al., 2022b).

*Table 18.* Comparison results between ∆-DiT and HarmoniCa on on MS-COCO for PIXART-$\alpha$ 1024 × 1024.

| Method | T | CLIP↑ | FID↓ | IS↑ | CUR(%)↑ | Speedup↑ |
|---|---|---|---|---|---|---|
| PIXART-$\alpha$ 1024 × 1024 (cfg = 4.5) | | | | | | |
| DPM-Solver++ (Lu et al., 2022b) | 20 | 31.07 | 31.98 | 41.30 | - | - |
| DPM-Solver++ (Lu et al., 2022b) | 13 | 31.04 | 33.29 | 39.15 | - | 1.54× |
| ∆-DiT (Chen et al., 2024b) | 20 | 30.40 | 35.88 | 32.22 | 37.49 | 1.49× |
| HarmoniCa ($\beta = 1e^{-3}$) | 20 | **31.05** | **32.53** | **40.36** | **59.31** | **1.73×** |

## J. Comparison between Learning-to-Cache and HarmoniCa with Different Sampling Strategies

For the implementation details [11], Learning-to-Cache uniformly samples an even timestep $t$ during each training iteration [12], as opposed to sampling any timestep from the set $\{1, \ldots, T\}$ as mentioned in Alg. 1 of its original paper. Consequently, according to Fig. 3, only $r_{t,i}$, where $t$ is an odd timestep, is learnable, while the remaining values are set to one. We compare Learning-to-Cache under different sampling strategies (*i.e.*, sampling an even timestep or without this constraint for each training iteration) against HarmoniCa. As shown in Tab. 19, our framework—whether training the entire Router or only parts of it (similar to the Learning-to-Cache implementation)—consistently outperforms Learning-to-Cache regardless of the sampling strategy.

It should be noted that the experiments in Sec. 5, with the exception of those in Tab. 9, use an implementation that uniformly samples an even timestep $t$ during each training iteration. This approach achieves significantly higher performance compared to sampling without constraints.

*Table 19.* Comparison results between Learning-to-Cache with different sampling strategies and HarmoniCa for the DiT-XL/2 $256 \times 256$. "♣" denotes that only parts of the `Router` corresponding to odd timesteps are learnable and the remaining values are set to one (*i.e.*, disable reusing cached features).

| Method | T | IS↑ | FID↓ | sFID↓ | Prec.↑ | Recall↑ | CUR(%)↑ | Latency(s)↓ |
|---|---|---|---|---|---|---|---|---|
| | | | | DiT-XL/2 $256 \times 256$ (`cfg` = 1.5) | | | | |
| DDIM (Song et al., 2020a) | 20 | 224.37 | 3.52 | 4.96 | 78.47 | 58.33 | - | 0.658 |
| Learning-to-Cache (Ma et al., 2024a) | 20 | 115.00 | 18.57 | 16.18 | 60.35 | 62.98 | 32.68 | $0.483_{(1.36\times)}$ |
| Learning-to-Cache♣ (Ma et al., 2024a) | 20 | 201.37 | 5.34 | 6.36 | 75.04 | 56.09 | 35.60 | $0.468_{(1.41\times)}$ |
| HarmoniCa♣ ($\beta = 3.5e^{-8}$) | 20 | 205.39 | **4.86** | 5.92 | 75.06 | 57.97 | 36.07 | $0.463_{(1.42\times)}$ |
| HarmoniCa | 20 | **206.57** | 4.88 | **5.91** | **75.20** | **58.74** | **37.50** | $\mathbf{0.456}_{(1.44\times)}$ |

*Table 20.* Accelerating image generation on MJHQ-30K (Li et al., 2024a) and sDCI (Urbanek et al., 2024) for the PIXART-$\alpha$. We sample 30K images for MJHQ-30K and 5K images for sDCI. "IR" denotes Image Reward.

| Method | T | MJHQ | | | | | sDCI | | | | | Latency (s)↓ |
|---|---|---|---|---|---|---|---|---|---|---|---|---|
| | | Quality | | | Similarity | | Quality | | | Similarity | | |
| | | FID↓ | IR↑ | CLIP↑ | LPIPS↓ | PSNR↑ | FID↓ | IR↑ | CLIP↑ | LPIPS↓ | PSNR↑ | |
| | | | | PIXART-$\alpha$ $512 \times 512$ (`cfg` = 4.5) | | | | | | | | |
| DPM-Solver++(Lu et al., 2022b) | 20 | 7.04 | 0.947 | 26.04 | - | - | 11.47 | 0.994 | 25.22 | - | - | 1.759 |
| DPM-Solver++(Lu et al., 2022b) | 15 | 7.45 | 0.899 | **26.02** | 0.138 | 21.41 | 11.55 | 0.876 | 25.19 | 0.178 | 19.85 | $1.291_{(1.36\times)}$ |
| HarmoniCa | 20 | **7.23** | **0.944** | 26.02 | **0.130** | **21.98** | **11.52** | **0.933** | **25.20** | **0.175** | **19.91** | $\mathbf{1.072}_{(1.64\times)}$ |
| | | | | PIXART-$\alpha$ $1024 \times 1024$ (`cfg` = 4.5) | | | | | | | | |
| DPM-Solver++(Lu et al., 2022b) | 20 | 6.24 | 0.966 | 26.23 | - | - | 10.96 | 0.986 | 25.56 | - | - | 9.470 |
| DPM-Solver++(Lu et al., 2022b) | 15 | 6.49 | 0.921 | 26.18 | 0.107 | 23.98 | 11.22 | 0.942 | 25.51 | 0.186 | 18.44 | $7.141_{(1.32\times)}$ |
| HarmoniCa | 20 | **6.40** | **0.931** | **26.19** | **0.102** | **25.06** | **10.99** | **0.969** | **25.53** | **0.183** | **20.23** | $\mathbf{5.786}_{(1.63\times)}$ |

## K. Results of T2I Generation on Additional Datasets and Metrics

In addition to the evaluations on ImageNet and MS-COCO, we conducted further tests using the high-quality MJHQ-30K (Li et al., 2024a) and sDCI (Urbanek et al., 2024) datasets with PIXART-$\alpha$ models. We added several metrics, including Image Reward (Xu et al., 2024), LPIPS (Learned Perceptual Image Patch Similarity) (Zhang et al., 2018), and PSNR (Peak Signal-to-Noise Ratio). The results, summarized in Tab. 20, demonstrate that HarmoniCa consistently outperforms DPM-Solver across all metrics on both the MJHQ and sDCI datasets. For instance, at the $512 \times 512$ resolution, HarmoniCa achieves an FID of 7.23 on the MJHQ dataset, which is lower than the 7.45 FID of DPM-Solver with 15 steps, indicating better image quality. Additionally, under the same configuration, HarmoniCa achieves a PSNR of 21.98, compared to DPM-Solver's 21.41 with 15 steps, reflecting better numerical similarity.

## L. Apply the Trained `Router` to a Different Sampler from Training During Inference

As shown in Tab. 21, the `Router` trained with one diffusion sampler can indeed be applied to a different sampler, such as DPM-Solver++→SA-Solver (6th row) and IDDPM→DPM-Solver++ (10th row). However, the performance of these trials is much worse than the standard HarmoniCa. We believe this is due to the discrepancies in sampling trajectories and noise scheduling between the two samplers, which need to be accounted for during the `Router` training. In other words, the sampler used for training should match the one used during inference to improve the performance.

## M. Results with Different Seeds

We conducted five independent runs with different random seeds on the following setting: PIXART-$\alpha$ $256 \times 256$ (Chen et al., 2023), using DPM-Solver++ (Lu et al., 2022b) (20 steps) and evaluating on 5000 images. The results in Tab. 22 show high consistency across runs, and more importantly, our caching-accelerated models consistently outperform the non-accelerated models in all evaluation metrics, confirming our method's robustness and effectiveness.

---

[10]$\Delta$-DiT presents the speedup ratio based on multiply-accumulate operates (MACs). Here we report the results according to the latency in that study.

[11]Let $T$ be an even number here.

[12]https://github.com/horseee/learning-to-cache/blob/main/DiT/train_router.py#L244-L247

*Table 21.* Results of applying the trained `Router` to a different sampler from training during inference. "A→B" denotes the `Router` trained with the sampler "A" is directly used during inference with the sampler "B".

| Method | T | CLIP↑ | FID↓ | sFID↓ | Latency(s)↓ |
|---|---|---|---|---|---|
| PIXART-α 256 × 256 (`cfg` = 4.5) | | | | | |
| SA-Solver (Xue et al., 2024) | 20 | 31.23 | 27.45 | 39.01 | 0.665 |
| SA-Solver (Xue et al., 2024) | 15 | 31.16 | 28.74 | 40.15 | $0.483_{(1.38\times)}$ |
| HarmoniCa | 20 | **31.20** | **27.98** | **39.26** | $\mathbf{0.420}_{(\mathbf{1.58}\times)}$ |
| HarmoniCa (DPM-Solver++→ SA-Solver) | 20 | 31.18 | 28.56 | 40.01 | $0.431_{(1.54\times)}$ |
| DPM-Solver++ (Lu et al., 2022b) | 100 | 31.30 | 25.01 | 35.42 | 2.701 |
| DPM-Solver++ (Lu et al., 2022b) | 73 | 31.27 | 25.16 | 36.11 | $2.005_{(1.35\times)}$ |
| HarmoniCa | 100 | **31.29** | **25.00** | **35.38** | $\mathbf{1.637}_{(\mathbf{1.65}\times)}$ |
| HarmoniCa (IDDPM→DPM-Solver++) | 100 | 31.24 | 26.11 | 40.24 | $1.648_{(1.64\times)}$ |

*Table 22.* Results with different seeds."*x/y*" in the table denotes the results that come from non-accelerated models and HarmoniCa. We use the same settings in Tab. 2 in the main text.

| Seed | IS↑ | FID↓ | sFID↓ |
|---|---|---|---|
| 8 | 33.28/33.27 | 37.31/35.44 | 94.78/92.10 |
| 16 | 33.61/33.62 | 37.43/35.46 | 94.74/92.11 |
| 24 | 33.59/33.60 | 37.55/35.48 | 95.01/92.32 |
| 32 | 33.65/33.65 | 37.11/35.34 | 94.87/92.13 |
| 40 | 33.64/33.64 | 37.05/35.29 | 94.88/92.19 |
| deviation | 0.138/0.144 | 0.188/0.073 | 0.093/0.081 |

## N. Qualitative Comparison and Analyses

As shown in Fig. 11 and 12, we provide qualitative comparison between HarmoniCa and other baselines, *e.g.*, Learning-to-Cache (Ma et al., 2024a), FORA (Selvaraju et al., 2024), and the fewer-step sampler. Our HarmoniCa with a higher speedup ratio can generate more accurate details, *e.g., 2nd column of Fig. 12 (d) vs. (b)* and objective-level traits, *e.g., 2nd column of Fig. 11 (d) vs. (c).*

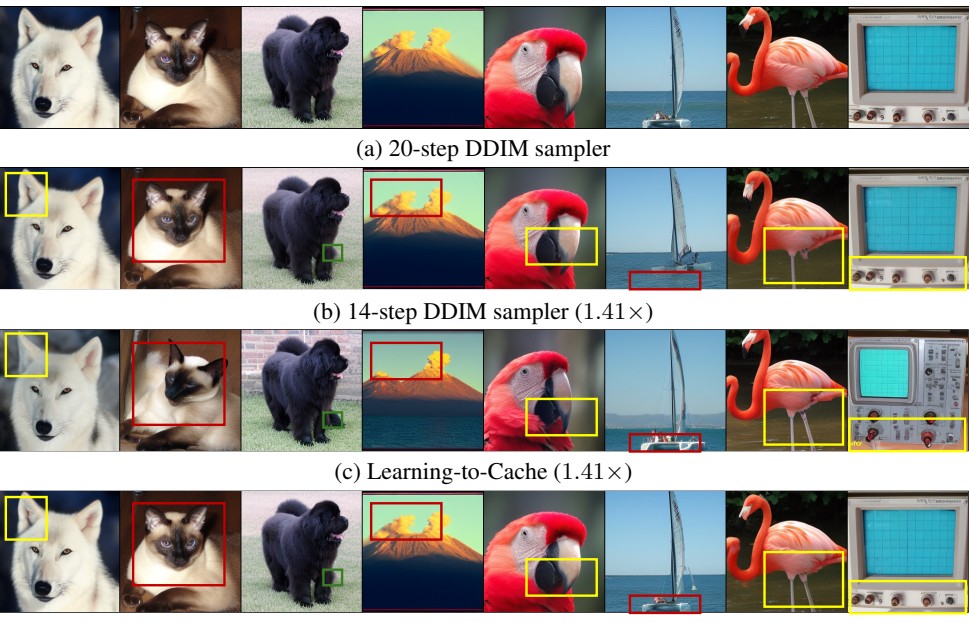

(a) 20-step DDIM sampler

(b) 14-step DDIM sampler (1.41×)

(c) Learning-to-Cache (1.41×)

(d) HarmoniCa (1.44×)

*Figure 11.* Random samples from DiT-XL/2 256 × 256 (Chen et al., 2023) with different acceleration methods. The resolution of each sample is 256 × 256. We employ `cfg = 4` here for better visual results. Key differences are highlighted using rectangles with various colors.

## O. Visualization Results

As demonstrated in Figures 13 to 16, we present random samples from both the non-accelerated DiT models and ones equipped with HarmoniCa, using a fixed random seed. We employ the same settings as those in the main text or more *aggressive* caching strategies. Our approach not only significantly accelerates inference but also produces results that closely resemble those of the original model. For a detailed comparison, zoom in to closely examine the relevant images.

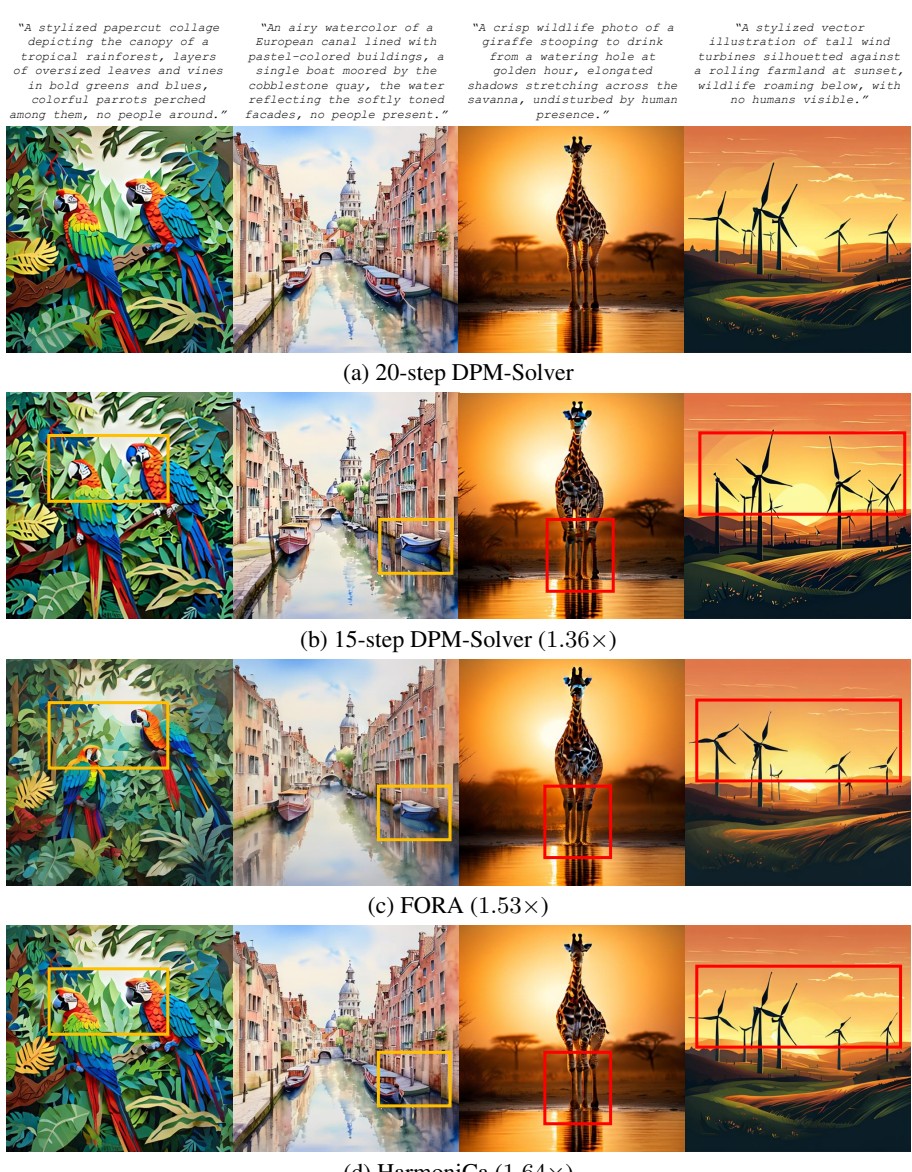

*"A stylized papercut collage depicting the canopy of a tropical rainforest, layers of oversized leaves and vines in bold greens and blues, colorful parrots perched among them, no people around."*

*"An airy watercolor of a European canal lined with pastel-colored buildings, a single boat moored by the cobblestone quay, the water reflecting the softly toned facades, no people present."*

*"A crisp wildlife photo of a giraffe stooping to drink from a watering hole at golden hour, elongated shadows stretching across the savanna, undisturbed by human presence."*

*"A stylized vector illustration of tall wind turbines silhouetted against a rolling farmland at sunset, wildlife roaming below, with no humans visible."*

(a) 20-step DPM-Solver

(b) 15-step DPM-Solver (1.36×)

(c) FORA (1.53×)

(d) HarmoniCa (1.64×)

*Figure 12.* Random samples from PIXART-α 512 × 512 (Chen et al., 2023) with different acceleration methods. The resolution of each sample is 512 × 512.

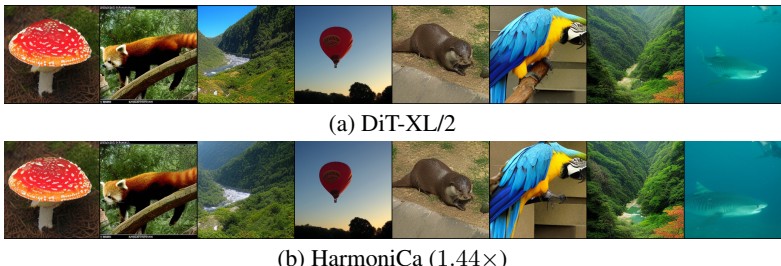

(a) DiT-XL/2

(b) HarmoniCa (1.44×)

*Figure 13.* Random samples from (a) non-accelerated and (b) accelerated DiT-XL/2 256 × 256 (Chen et al., 2023) with a 20-step DDIM sampler (Song et al., 2020a). The resolution of each sample is 256 × 256. We mark the speedup ratio in the brackets.

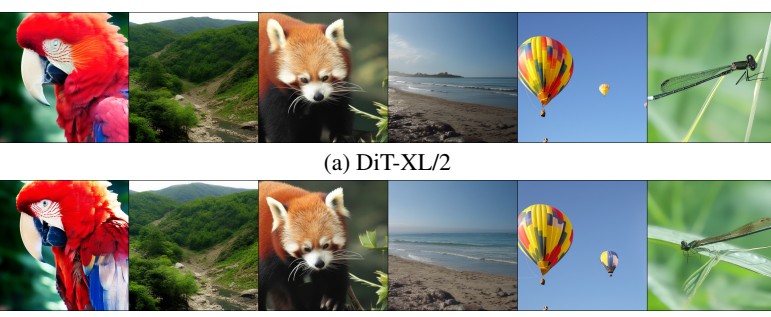

(a) DiT-XL/2

(b) HarmoniCa (1.30×)

*Figure 14.* Random samples from (a) non-accelerated and (b) accelerated DiT-XL/2 $512 \times 512$ (Chen et al., 2023) with a 20-step DDIM sampler (Song et al., 2020a). The resolution of each sample is $512 \times 512$.

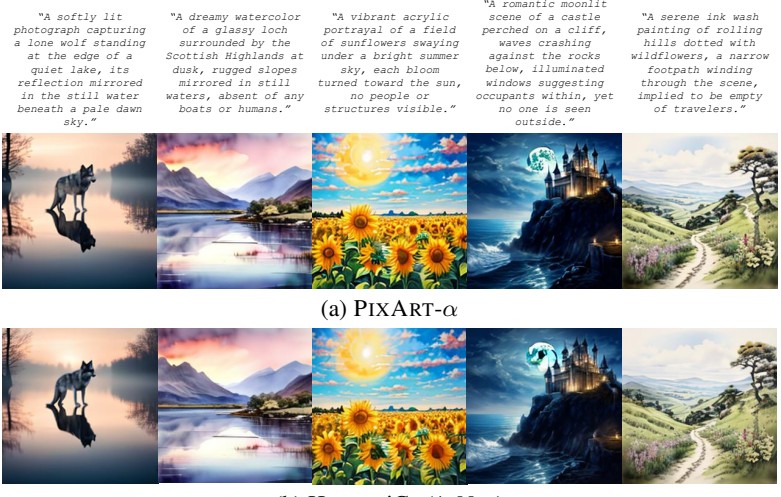

(a) PIXART-$\alpha$

(b) HarmoniCa (1.60×)

*Figure 15.* Random samples from (a) non-accelerated and (b) accelerated PIXART-$\alpha$ $256 \times 256$ (Chen et al., 2023) with a 20-step DPM-Solver++ sampler (Lu et al., 2022b). The resolution of each sample is $256 \times 256$. Text prompts are exhibited above the corresponding images

*"A minimalist watercolor composition featuring a single, lonely tree atop a gentle hill, its delicate branches set against a soft gradient sky, with no surrounding figures or objects."*

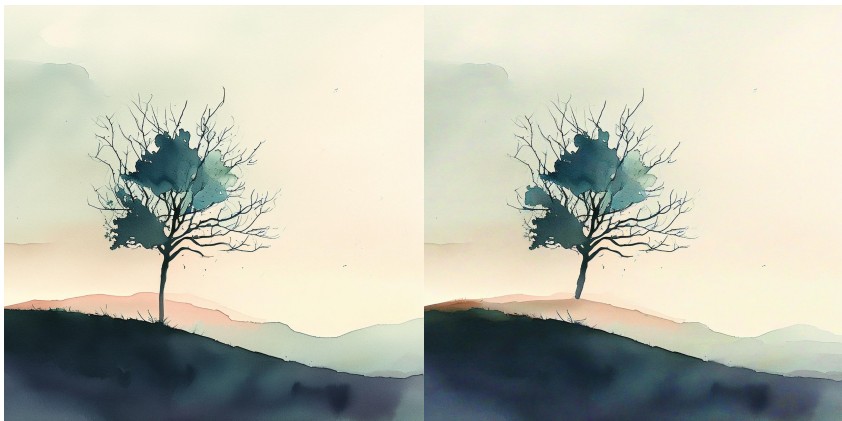

*"An expressive oil painting of a secluded cabin by a winding river, tall pines reflecting on the water's surface, a faint glow from the cabin windows hinting at life within, yet no one is shown outside."*

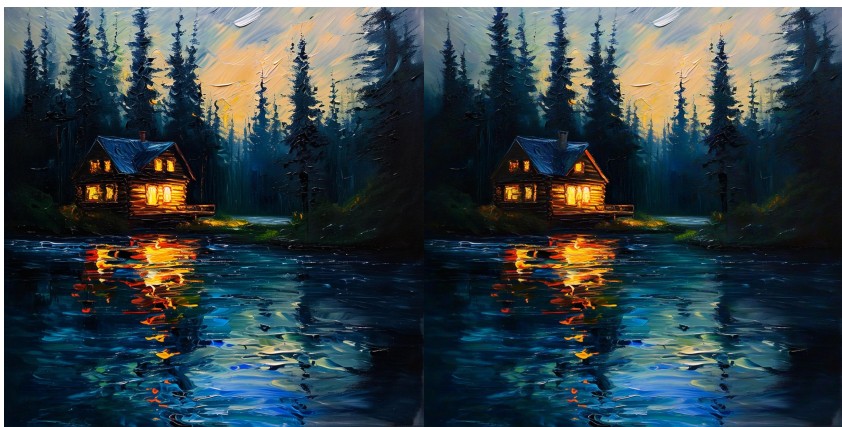

*"An acrylic painting of a tranquil lagoon filled with lily pads and reed-fringed shores, warm twilight hues reflecting across the water, absent of any human presence."*

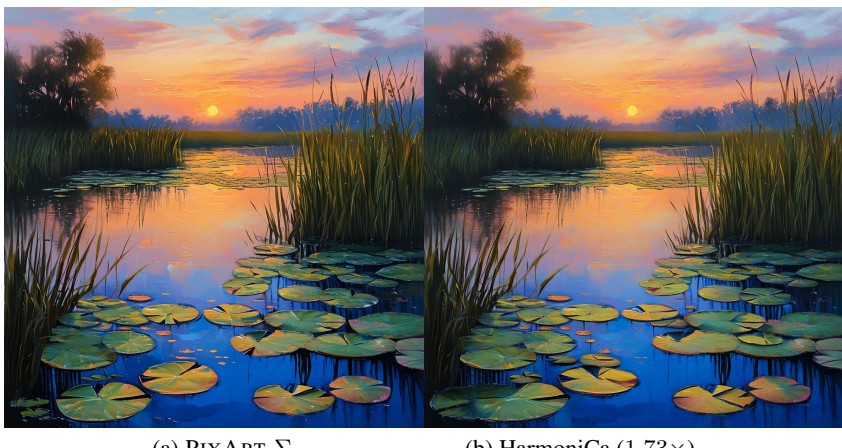

(a) PIXART-Σ (b) HarmoniCa (1.73×)

*Figure 16.* Random samples from (Left) non-accelerated and (Right) accelerated PIXART-Σ-2K (Chen et al., 2024a) with a 20-step DPM-Solver++ sampler (Lu et al., 2022b). The resolution of each sample is 2048 × 2048.

