# OpenReview forum: "HarmoniCa: Harmonizing Training and Inference for Better Feature Caching in Diffusion Transformer Acceleration"
_ICML.cc/2025/Conference — ICML 2025 poster_

### Official Review · Reviewer_5nDy · 2025-03-11

**Overall Recommendation:** 1

**Summary:**

This study introduces a novel caching framework to enhance inference efficiency in diffusion transformers. The core idea is to employ a learnable router that determines optimal caching dynamics during the generation process. The primary contribution lies in developing a new learning paradigm that alleviate the error accumulation in previous method (which utilizes a skip-step training strategy). Besides, a dynamic loss reweights the training objective of the proposed learning paradigm. Guided by final output quality, the reweighting strategy, as the authors stated, can minimizing the discrepancies between cached and non-cached generateion pathways.

## Update After Rebuttal

The rebuttal clarified the authors’ perspective on the methodological novelty of the proposed SDT and IEPO training strategies, which they position as innovations. However, key disagreements persist regarding the substantiation of these claims. My primary concerns are as follows:

1. **Questionable novelty of SDT/IEPO**:​​ The empirical results do not convincingly demonstrate that the performance gains stem uniquely from SDT/IEPO rather than the baseline L2C framework. For instance, the reported acceleration improvements show negligible divergence from those achieved by L2C alone, suggesting that the observed benefits may originate from the underlying L2C methodology rather than the proposed strategies.

2. **Insufficient empirical validation of methodological impact**: The rebuttal’s discussion of teacher forcing—where the authors acknowledge that strict teacher forcing yields comparable results to their proposed strategy—fens the case for SDT’s significance. If varying training strategies (including their own) produce similar outcomes, the necessity and distinct contribution of SDT remain unsubstantiated.

Besides, I would like further clarify my **score justification** in the rebuttal: I mean **either (not both)** technical innovation or theoretical breakthrough is required for an ICML paper. This paper mainly provides technical innovation which should be rigorously validated through measurable performance gains or demonstrate methodological superiority. In this work, neither criterion is met: the strategies lack clear differentiation from existing approaches (e.g., L2C), and the marginal empirical improvements fail to justify their novelty. (**That is also why I am suggesting exploring theoretical foundations**: I find that it may be hard to explain why the gains compared to L2C is marginal.)

Overall, the paper does not meet the threshold for technical novelty required for ICML. Thus, I maintain my recommendation for rejection.

**Claims And Evidence:**

Somewhat convincing. Though experimental results exhibit overall marginal performance boost, I still do not believe that the proposed strategy (SDT and IEPO) can solve the error accumulation problem. (See Other Strength and Weakness)

**Essential References Not Discussed:**

While the submission adequately cites foundational works in caching-based method (e.g. learn2cache), some merging-based works are missing such as token merge. (See Other Strength and Weakness)

**Experimental Designs Or Analyses:**

While the experimental framewok adopts conventional benchmark, there are some concerns about the efficacy of the proposed method. (See Other Strength and Weakness)

**Methods And Evaluation Criteria:**

Somewhat convincing.

**Other Comments Or Suggestions:**

See Weaknesses.

**Other Strengths And Weaknesses:**

## Strengths

The topic is very practical and enjoys a wide range of appications.

## Weaknesses

1. (Minor) Literature Review and Citations

+ **Some key works are missing.** The discussion overlooks recent advancements in token merging (which is also an important method in efficient diffusion model), such as ToMe [1], TokenFusion [2], and attention-driven efficiency method [3].

+ **Citation.** Multiple references cite preprint versions despite peer-reviewed alternatives being available. Please update these to converence/journal citations to strengthen scholarly rigor.

2. (Major) Experiments

+ **Limited Baseline.** Some baselines should be included such as $\Delta$-DiT [4]. Besides, leveraging a router to determine the inference caching pathway is not a fresh idea. Thus, [5] is also suggested as a baseline. (This is optional as it is publicly avaiable within 4 mounths. According to the Reviewer Instructions, it is not a concurrent work.)

+ **Performance Significance.** The reported latency improvements (less than 0.1s for most experiments) lack practical significance, particularly for application-oriented claims.

+ **Limitations and Failure Cases.** The manuscript omits scenarios where the method underperforms. Including qualitative examples of failure modes would clarify its operational boundaries and foster reproducibility, providing valuable insights on the scope of application of the proposed method.

3. (Major) Methodology

+ **Novelty Concerns.** The core methodology appears incremental, building primarily on Learn2Cache with refinements (i.e. the SDT and IEPO). However, these caching-based methods are quite crowded. A sufficient efforts have been made to explore the layerwise redundancy. Given this, this paper does not provide clear insight to the community.

+ **Generazation Risks.** (not for this paper only, but for all training-based methods) The router’s sample-specific computation flow raises questions about robustness. For example, could certain inputs bypass acceleration entirely due to routing decisions? Empirical validation on out-of-distribution or edge-case samples is absent.

+ **Theoretical Foundations.** The theoretical justification for SDT’s error reduction remains unclear. For instance, while SDT purportedly stabilizes training invariance, its dependency on cached features (ε') risks error propagation—a concern unaddressed in the current analysis. (Maybe a teacher-forcing like strategy is better?)

[1] Daniel Bolya and Judy Hoffman. Token merging for faststable diffusion. CVPR 2023

[2] Kim et. al. Token fusion: Bridging the gap between token pruning and token merging. WACV 2024

[3] Wang el. al. Attention-driven training-free efficiency enhancement of diffusion models. CVPR 2024

[4] Chen et. al. $\Delta $-DiT: A Training-Free Acceleration Method Tailored for Diffusion Transformers. Arxiv 2024.06

[5] You et. al. Layer- and Timestep-Adaptive Differentiable Token Compression Ratios for Efficient Diffusion Transformer. Arxiv 2024.12

**Questions For Authors:**

+ What is the prompt dataset utilized in Figure 12, 15 and 16?

+ What is the architecture of the router? Is this architecture identical to the one described in learn2cache? If so, more detailed configurations are expected. If not, the extra computational overhead should be quantified and provided.

**Relation To Broader Scientific Literature:**

The key contributions of this paper (SDT and IEPO) can be seen as adjustments to the Learn2Cache. Learn2cache is a sample-specific caching method that alters the inference caching pathway according to each generated samples.

**Theoretical Claims:**

N/A. Providing formal analysis will significantly strengthen the theoretical foundations (while theoretical validation is not explicitly addressed). I find some directions that may help strengthen the theoretical foundations:

+ The error. Analysis the error propagation and quantify the error accumulation rate and demonstrate why SDT is better than skip-step strategy.
+ The reweighting strategy. Analysis on the convergence dynamics of reweighting strategy, especailly how this strategy alters the convergence properties of Eq.4.

---

> ### Author Rebuttal · Authors · 2025-04-01
>
> Thanks to the reviewer for the constructive comments.
>
> **Literature Review and Citations**
>
> - **Missing work.** We appreciate the suggestions and will include ToMe, TokenFusion, and attention-driven efficiency methods in the revision.
> - **Citation updates.** We will replace arXiv references with conference/journal versions where available.
>
> **Experiments**
>
> - **Baselines.** We have already included $\Delta$-DiT in Sec. H, showing our method outperforms it in both accuracy and efficiency. For [1], it was accepted by CVPR after the ICML deadline and appeared on arXiv after Oct. 2024. According to [ICML policy](https://icml.cc/Conferences/2025/ReviewerInstructions), it qualifies as a concurrent work. Moreover, its code is not available, making comparison infeasible within the rebuttal window.
> - **Performance Significance.** We respectfully disagree since a larger batch or model size can show larger absolute speedup. While some latency results are based on small models (e.g., DiT-XL/2 with batch size 8) for controlled ablations or comparisons, they still show consistent speedups without sacrificing quality (e.g., Tab. 2). On larger models like PixArt-$\Sigma$ $2048\times2048$, our method yields much larger absolute gains and higher acceleration ratios (Sec. E).
> - **Failure Cases.** Across all tested settings (i.e., models, datasets, and samplers), our method shows robust and consistent performance. We did not observe clear failure cases that warrant separate analysis. Moreover, such analysis is uncommon in related works and not part of standard practice in this field.
>
> **Methods**
>
> - **Novelty Concerns.** While our work builds upon the foundational concept (i.e., learn a caching strategy) introduced by Learning-to-Cache (L2C), our contributions and improvements are both significant and distinct:
>     1. **Identifying Key Problems**: We first identify two critical discrepancies between training and inference overlooked in L2C as follows：
>         1. **Prior timestep disregard**: Ignore the error caused by reusing cached features at prior timesteps during optimization.
>         2. **Objective Mismatch:** Focus on optimizing the intermediate output during the denoising process.
>
>         They both significantly affect the overall performance of L2C.
>
>     2. **Novel Training Strategies**: To address these problems, we propose two novel training techniques(i.e., SDT+IEPO). **Our entire optimization processes differ fundamentally** compared with L2C:
>         1. For the training paradigm, L2C uses random timestep sampling similar to DDPM, focusing solely on the current sampled timestep. In contrast, we implement SDT for iterative denoising from random noise, which accounts for all previous timesteps, mirroring the inference process.
>         2. For the learning objective, L2C focuses on intermediate noise and cache usage. We further optimize the final image error and cache usage with our IEPO.
>     3. **Reduced Training Cost**: For our method, **no training images** are needed, and **training time is much shorter** than L2C. This makes it applicable to large-scale models like PixArt.
>     4. **Substantial Performance Improvements:** Our method consistently outperforms L2C. For instance, Tab. 2 shows **+1.24** FID and **+6.74** IS on DiT-XL/2. Fig. 7 shows **+30.90** IS and **−12.34** FID at $1.6\times$ speedup.
> - **Generalization.** Our router is static once trained and does not change per sample during inference. The method is also image-free during training without tuning the weights, reducing the risk of overfitting. Moreover, we can observe strong generalization across multiple datasets and model sizes in this paper.
> - **Theoretical Foundation.** The reviewer suggests using a teacher-forcing strategy—this is exactly what SDT employs during training. Our experiments (e.g., Fig. 5, Tab. 9) validate its effectiveness in reducing error accumulation. A more detailed theoretical analysis is planned for future work, as we mainly focus on practical speedup while maintaining the performance in this work.
>
> **Questions for Authors**
>
> - Prompts in Figs. 12, 15, and 16 are randomly generated using GPT-4.
> - The router structure is identical to L2C, with all the details in Sec. 5.1 and Sec. D.
>
> [1] You et. al. Layer- and Timestep-Adaptive Differentiable Token Compression Ratios for Efficient Diffusion Transformer. Arxiv 2024.12

---

> > ### Comment · Reviewer_5nDy · 2025-04-02
> >
> > Thanks for your rebuttal.
> >
> > ### **Experiments---Performance Significance**
> >
> > I would like to clarify that the limited improvements refer to the marginal improvement over L2C (as evidenced by Table 2's acceleration metrics) given that the proposed method builds upon L2C as its foundation. Besides, in some experiments, the L2C is not even included as a baseline. It is skeptical whether the acceleration is brought by L2C or the proposed SDT+IEPO.
> >
> > ### **Methods---Generalization**
> >
> > What I mean is that the training process requires condition (such as text) as input. For example, for text-to-image models, the cross-prompt generalization is a concern.
> >
> > ### **Methods---Theoretical Foundation**
> >
> > I do not agree that the proposed method leverages a teacher-forcing strategy: $x_t$ is updated by the noise $\epsilon^{(t)'}$ (rather than $\epsilon^{(t)}$) which is produced by the **cached model** (as mentioned in Line 13 of Alg. 1). Potential error accumulation still occurs from iterative cache-dependent noise prediction and update.
> >
> > ### **Score Justification**
> >
> > While I acknowledge the authors' efforts to address review concerns through their rebuttal, I keep my original score considering the paper's suitability for ICML's high standards: Given that ICML's emphasis on both algorithmic innovation and theoretical rigor, the current submission neither sufficiently advance beyond incremental engineering improvements nor provide mathematical depth required for a machine learning theory conference.

---

> > > ### Author Response · Authors · 2025-04-03
> > >
> > > Thank you for your thoughtful comments. We would like to address the concerns raised in your latest response.
> > >
> > > - **Performance Significance.** Firstly, we would like to emphasize that in Tab. 2, we **control the acceleration ratio of HarmoniCa, which is **not built upon Learning-to-Cache (see the end of this paragraph)**, slightly higher than baselines to show the performance improvement**, which of HarmoniCa over L2C is **non-trivial** and **significant**. In the official paper of L2C, also with a slightly higher speedup ratio, the improvements of Learning-to-Cache beyond other caching methods (e.g., DeepCache and FasterCache) on FID are **all below 0.3** points (Tab. 3 in the official paper). However, our method, with significantly lower training cost (Sec. 4.3), achieves **1.24 FID improvement** over L2C (Tab. 2). Secondly, as we mentioned in Sec. 5.1, L2C requires the datasets in the pretrain stage for training, which requires significant costs. The authors of L2C also exclude text-to-image tasks in their paper, which means there’s no official code for these experiments. Further, considering that our method achieves consistently better performance for all metrics in Tab. 2, we believe our method has already shown superiority compared with L2C. Thus, it is reasonable not to include it in text-to-image experiments. Finally, the improvements in these experiments are definitely achieved by SDT+IEPO. **Specifically, instead of plug-and-play approaches that build upon L2C, SDT and IEPO are both more advanced training strategies, which are parallel and independent to the methods (i.e., training paradigm+loss function) proposed in L2C.**
> > > - **Generalization.** Thank you for your comment. Our router for the text-to-image model is trained with 1000 randomly picked COCO captions beyond the evaluation set. It not only performs well on the COCO evaluation set, but also on MJHQ and sDCI, as demonstrated in Sec. 9. Visualizations (prompts are randomly generated by GPT-4) without cherry-picking in the manuscripts can further validate that our method generalizes effectively beyond the prompts used for training.
> > > - **Theoretical Foundation.** Regarding your comment on teacher-forcing, we broadly believe that employing a pretrained model (i.e., teacher) to guide the learning process is teacher-forcing, which is slightly different from [wikipedia](https://en.wikipedia.org/wiki/Teacher_forcing). However, we also compare the teacher-forcing, using $\mathbf{\epsilon}^{(t)}$ from teacher to feed for the next iteration, with our HarmoniCa. As shown in the following table. HarmoniCa shows comparable results, which indicate that using  $\mathbf{\epsilon}^{(t)'}$ would not lead to potential error accumulation compared with  $\mathbf{\epsilon}^{(t)}$. A more detailed theoretical analysis is planned for future work, as we mainly focus on practical speedup while maintaining the performance in this work.
> > >
> > >     *The 2nd and 3rd row are for DiT-XL/2 (20 step) on ImageNet $256\times 256$, and others for DiT-XL/2 (20 steps) on ImageNet $512\times512$.*
> > >
> > >     | Method | FID$\downarrow$ | IS$\uparrow$ | sFID$\downarrow$ | Latency(s)$\downarrow$ | Speedup$\uparrow$ |
> > >     | --- | --- | --- | --- | --- | --- |
> > >     | HarmoniCa ($\mathbf{\epsilon}^{(t)'}$ ) | 4.88 | 206.57 | **5.91** | 0.456 | $1.44\times$ |
> > >     | Teacher-forcing ($\mathbf{\epsilon}^{(t)}$ ) | **4.87** | **207.12** | 6.02 | 0.458 | $1.44\times$ |
> > >     | HarmoniCa ($\mathbf{\epsilon}^{(t)'}$ ) | **5.72** | **179.84** | **6.61** | 2.574 | $1.30\times$ |
> > >     | Teacher-forcing ($\mathbf{\epsilon}^{(t)}$ ) | 5.74 | 178.96 | **6.61** | 2.577 | $1.30\times$ |
> > > - **Score Justification.** We appreciate the concerns, but disagree with the highest respect. Going far beyond engineering improvements, our work is the first to pinpoint important problems (discrepancies between training and inference under cache usage) in previous research and introduces innovative training strategies to effectively and efficiently solve them with extensive experiments. Moreover, as the conference specializes **Application$\rightarrow$Computer Vision** as one of the primary areas this year, we believe our application-driven work, not a theoretical paper, is aligned with ICML’s emphasis on practical advancements in machine learning and offers substantial improvements in real-world application scenarios.

---

### Official Review · Reviewer_u3jx · 2025-03-13

**Overall Recommendation:** 3

**Summary:**

This paper introduces HarmoniCa, a learning-based caching framework that addresses two key discrepancies between diffusion training and inference: ***prior timestep disregard*** and ***objective mismatch***. First, it presents Step-wise Denoising Training (SDT), which adopts a student-teacher setup to consider full denoising trajectories during diffusion training. The student employs a caching system to store the denoising trajectory, while the teacher leverages the student to indirectly incorporate this trajectory. Second, it introduces an Image Error Proxy-Guided Objective (IEPO) to supervise diffusion models to approximate the final image error, allowing SDT to balance cache usage and image quality. Extensive experiments verify the effectiveness of HarmoniCa.

**Claims And Evidence:**

The authors address the claims in ***prior timestep disregard*** and ***objective mismatch*** with the previously introduced caching framework [1]. However, their claims appear to be limited to diffusion models. Recently, several rectified flow models [2, 3, 4] have been proposed, which train on straight ordinary differential equation (ODE) paths that naturally mitigate these issues. In these models, the denoising trajectory follows a nearly linear interpolation between source and target distributions, leading to minimal error accumulation and interference. Given the complexity of both training and sampling pipelines compared to rectified flow models, HarmoniCa contributes only marginal improvements in diffusion models, and I am not fully convinced that their claims are well-substantiated.

Furthermore, the relationship between the discrepancy and the efficient caching system remains unclear. The authors have to provide a claim on this relationship.

[1] Ma et al., Learning-to-cache: Accelerating diffusion transformer via layer caching, NeurIPS 2024. \
[2] Liu et al., Flow Straight and Fast: Learning to Generate and Transfer Data with Rectified Flow, ICLR 2023. \
[3] Lee et al., Improving the Training of Rectified Flows, NeurIPS 2024. \
[4] Liu et al., InstaFlow: One Step is Enough for High-Quality Diffusion-Based Text-to-Image Generation, ICLR 2024.

**Essential References Not Discussed:**

N/A

**Experimental Designs Or Analyses:**

The authors provide in-depth analysis and various metrics in evaluating image quality and inference efficiency.

**Methods And Evaluation Criteria:**

The proposed methods are well-aligned with their claims, where SDT effectively handles the outputs from earlier timesteps as caches and IEPO makes SDT more efficient. Extensive experiments across various models, samplers, and image resolutions validate the effectiveness of HarmoniCa.

**Other Comments Or Suggestions:**

As HarmoniCa addresses a new training framework, the authors have to provide some analysis or experiments regarding its effectiveness compared to other powerful generative modeling, such as rectified flow models.

**Other Strengths And Weaknesses:**

* **Potential strengths.** I believe the comprehensive experiments in HarmoniCa demonstrate its high potential for application across various generative models.

* **Potential weakness.** However, it lacks a clear logical argument to support its extensive experimental findings. Moreover, the expSeveral hyper-parameters limit the adaptability of HarmoniCa.

**Questions For Authors:**

My concerns are summarized into:

1. The authors have to provide some analysis or experiments regarding its effectiveness compared to other powerful generative modeling, such as rectified flow models.

2. The relationship between the discrepancy and the efficient caching system remains unclear. The authors have to provide a claim on this relationship.

**Relation To Broader Scientific Literature:**

In my opinion, a caching system itself holds significant potential for application across various generative models. However, the concept of a learnable caching system was previously introduced in Learning-to-Cache. Moreover, this approach appears to offer only marginal improvements within diffusion models and does not seem applicable to recent rectified flow models.

**Theoretical Claims:**

No theoretical claims and proofs.

---

> ### Author Rebuttal · Authors · 2025-04-01
>
> Thanks for the reviewer’s constructive comments.
>
> > **Claims And Evidence**
> >
> - **Comparison with rectified flow models.** We clarify that our approach is not based on pretraining, unlike flow-based models. A direct comparison is not entirely appropriate, as our method merely trains a small tensor (i.e., the Router) and only focuses on improving inference efficiency in a post-training manner. Thus, we would like to emphasize that our methods can bring significant benefits compared with other post-training methods (e.g., quantization, pruning, and other caching methods in Sec. 5). Moreover, HarmoniCa can be effectively applied to rectified flow models. We employ the pretrained models and evaluation in [LFM](https://vinairesearch.github.io/LFM/). As shown in the following table ( *$T=100$*, Euler Solver, and $BS=8$), our method achieves substantial speedup ratios without performance degradation.
>
>
>     | Datasets | Models | FID$\downarrow$ | Latency (s)$\downarrow$ | Speedup$\uparrow$ |
>     | --- | --- | --- | --- | --- |
>     | LSUN-Bedroom $256\times256$ | DiT-L/2 | 5.22 | 1.76 | - |
>     | + HarmoniCa | DiT-L/2 | **5.13** | **1.09** | **$1.62\times$** |
>     | CelebaHQ $256\times256$ | DiT-L/2 | 5.42 | 1.76 | - |
>     | + HarmoniCa | DiT-L/2 | **5.41** | **1.07** | **$1.65\times$** |
>     | ImageNet $256\times256$ | DiT-B/2 (CFG=1.5) | 5.06 | 1.37 | - |
>     | + HarmoniCa | DiT-B/2 (CFG=1.5) | **5.04** | **0.87** | **$1.58\times$** |
> - **Relationship between the discrepancy and the caching system.** We further explain the relationship as below (also can be found in Sec. 4.1):
>     1. **Prior Timestep Disregard**: In previous research, such as Learning-to-Cache, the training process employs a raw image with manually added noise as input $\mathbf{x}_t$(see Fig. 3), which is unaffected by any cached feature reuse. However, during accelerated inference, the input (i.e., a noisy image) is disrupted by cached feature reuse in earlier timesteps (as illustrated in Lines 187-191). Moreover, the cache context in current timestep is shaped by the caching strategy in earlier timesteps (e.g., $t +1,\ldots, T$), which is unconsidered as shown in Fig. 3. This discrepency of disregarding the caching effects of prior timestep can lead to significant error, which is caused by cache usage, accumulation (see Fig. 5). To address this, SDT (Sec. 4.2) ensures that the model learns to incorporate the prior timestep's effect on the cache, aligning the training process with the inference stage and reducing error accumulation.
>     2. **Objective Mismatch**: Equipped with the caching strategy, the training objective in previous research typically minimizes each timestep noise prediction error caused by caching usage. However, inference aims to generate high-quality images, the same as those from non-accelerated models. This mismatch between the two processes leads to an optimization deviation (Sec. C) and image distortion (Fig. 6). By integrating IEPO (Sec. 4.2), we ensure that the training process considers the final image error, which mitigates the optimization shift and improves the quality of generated images.
>
> > **Relation To Broader Scientific Literature**
> >
>
> We appreciate the reviewer’s insight. While Learning-to-Cache introduced the concept, HarmoniCa extends this by addressing the discrepancies between training and inference, and also requires fewer training resources (Sec. 4.3).  Moreover, HarmoniCa achieves much higher efficiency and performance across all metrics compared with previous methods, including caching, quantization, and pruning. Regarding the concern about the marginal improvements, we acknowledge that the gains in some small models may appear modest. However, the benefits become more pronounced in large-scale models (e.g., Figs. 3 and 12). Besides, we emphasize that HarmoniCa can be effectively applied to rectified flow models (see **Claims And Evidence**).
>
> > **Other Strengths And Weaknesses**
> >
> - **Lacks a clear logical argument.** Regarding the concern, we clarify our general and natural logical argument process as follows. First, we have carefully analyzed the two discrepancies between training and inference. Then, we proposed our framework to solve the problems. Both parts include detailed logical analyses and experimental validation. These processes provide the rationality to support the large number of experimental findings in Sec. 5. We will make the paper logic clearer in the revision.
> - **Hyperparameters limit the adaptability.** We would like to clarify that we have three hyperparameters in total: interval $C$, threshold $\tau$, and regularization coefficient $\beta$. The superior performance and speedup across all experiments validate the efficacy of using fixed $C$ and $\tau$. Therefore, the only one that requires adjustment is $\beta$, which determines the trade-off between acceleration and image quality. We will explore the automatic methods to determine $\beta$ in the future.

---

> > ### Comment · Reviewer_u3jx · 2025-04-03
> >
> > Thanks for the clarification. It's interesting that HarmoniCa can improve both FID and latency in LFM.
> >
> > Additionally, can you provide the individual effectiveness of SDT and IEPO, respectively? Based on the performance of Learning-to-Cache in Table 2 and the ablation results in Table 9, the effectiveness of IEPO is evident. However, it’s difficult to isolate the contribution of SDT alone. Is SDT only effective when combined with specific learning objectives? I think it seems difficult to find analyses of SDT.

---

> > > ### Author Response · Authors · 2025-04-04
> > >
> > > Thanks for your constructive feedback.
> > >
> > > Considering the training paradigm and objective of Learning-to-Cache (L2C) as the naive strategies, the individual effectiveness of SDT and IEPO can be found in *the 4th row vs. the 6th row* and *the 4th row vs. the 5th row* in Tab. 9, respectively. For SDT, we believe the improvement is noticeable. Specifically, it boosts 10.56 (w/ $\mathcal{L}\_{LTC}^{(t)}$) and 0.32 FID (w/ $\mathcal{L}_{IEPO}^{(t)}$) enhancements compared with the training paradigm of L2C. Since L2C+ $\mathcal{L}\_{IEPO}^{(t)}$ has already achieved a new state-of-the-art FID score of 5.20, we would like to emphasize that the improvement of SDT is non-trivial when using $\mathcal{L}\_{IEPO}^{(t)}$.  Moreover, it is also worthwhile to highlight that SDT only requires significantly less training time under the same settings (e.g., batch size and iterations) without any data here compared with the training paradigm of L2C (Sec. 4.3).

---

### Official Review · Reviewer_CT5R · 2025-03-13

**Overall Recommendation:** 3

**Summary:**

This paper introduces a method for improving the cache mechanism in diffusion transformers. The proposed method including two parts:
* A step-wise denoising training. It changes the objective in Learning-to-cache from single step to a all step objective.
* An image error proxy-guided objective. The authors propose to use the matching of the final image as an extra guidance for the caching router.

Experiments show that it surpasses Learning-to-Cache and FORA at a high CUR setting. The authors also conduct extensive analysis experiments to compare their results with diffusion compression and sampling methods.

**Claims And Evidence:**

The claim and the motivation are correct, but I believe the experimental validation is insufficient to fully support their claims. Please refer to section Experimental Designs or Analyses.

**Essential References Not Discussed:**

N/A

**Experimental Designs Or Analyses:**

(1) The performance improvement of the proposed method over learning-to-cache is quite limited.

In the low CUR setting (Table 13), there is almost no performance gain, with improvements only observed in high CUR scenarios. Moreover, the results presented in the tables do not match those reported in the original papers. In Table 2, for example, at 50 timesteps, the original paper reports **an FID of 2.27 with a 1.3× speedup**, while the authors report Learning-to-cache result with **an FID of 2.62 with a 1.25× speedup**. The proposed method here achieves **an FID of 2.36 with a 1.3× speedup**, which is even worse than the results in learning-to-cache. For 20 timesteps, Figure 4 in the learning-to-cache paper shows that at a 1.4× speedup, the FID is below 4.5. However, the authors report an FID of 5.34 for learning-to-cache, while this method achieves 4.88.

(2) The results in the ablation study are inconsistent with Table 2.

In Table 9, the authors report that the learning-to-cache with $L_{LTC}$ achieves an FID of 18.57 (If I understand correctly this would be the same as the algorithm of learning-to-cache), which is inconsistent with results for learning-to-cache in Table 2 (FID = 5.34). It is unclear what the authors are basing their claim on Line 387, where they state that their method (SDT) achieves an FID improvement of 10 over the learning-to-cache.

**Methods And Evaluation Criteria:**

Yes, the method proposed by the authors is reasonable, but it is incremental compared to learning-to-cache.

The motivation stated in the paper is to address two inconsistencies between training and testing in the design of learning-to-cache algorithms: (1) only considering a single step and (2) optimizing for the noise in a single-step prediction rather than the final image.

The proposed solution can be broadly summarized as transforming the single-step learning-to-cache approach into an optimization objective that considers the entire denoising process, taking into account how cached results from earlier steps affect later steps. This involves moving from sampling single steps, as in learning-to-cache, to running a full denoising process, computing the loss at each step, performing backpropagation for each loss, and incorporating a proxy loss derived from the final image.

I find this contribution insufficiently novel.

**Other Comments Or Suggestions:**

N/A

**Other Strengths And Weaknesses:**

My main concern is the novelty and the experimental design of the method. Please refer to previous section.

**Questions For Authors:**

None

**Relation To Broader Scientific Literature:**

An improved method for learn-to-cache

**Theoretical Claims:**

N/A

---

> ### Author Rebuttal · Authors · 2025-04-01
>
> Thanks for the reviewer’s constructive comments.
>
> > **Methods And Evaluation Criteria**
> >
>
> **Contribution insufficiently novel.** While our work builds upon the foundational concept (i.e., learn a caching strategy) introduced by Learning-to-Cache (L2C), our contributions and improvements are both significant and distinct:
> 1. **Identifying Key Problems**: We first identify two critical discrepancies between training and inference overlooked in L2C as follows：
>     1. **Prior timestep disregard**: Ignore the error caused by reusing cached features at prior timesteps during optimization.
>      2. **Objective Mismatch:** Focus on optimizing the intermediate output during the denoising process.
>
>      They both significantly affect the overall performance of L2C.
>
> 2. **Novel Training Strategies**: To address these problems, we propose two novel training techniques(i.e., SDT+IEPO). **Our entire optimization processes differ fundamentally** compared with L2C:
>      1. For the training paradigm, L2C uses random timestep sampling similar to DDPM, focusing solely on the current sampled timestep. In contrast, we implement SDT for iterative denoising from random noise, which accounts for all previous timesteps, mirroring the inference process.
>       2. For the learning objective, L2C focuses on intermediate noise and cache usage. We further optimize the final image error and cache usage with our IEPO.
> 3. **Reduced Training Cost**: For our method, **no training images** are needed, and **training time is much shorter** than L2C. This makes it applicable to large-scale models like PixArt.
> 4. **Substantial Performance Improvements:** Our method consistently outperforms L2C. For instance, Tab. 2 shows **+1.24** FID and **+6.74** IS on DiT-XL/2. Fig. 7 shows **+30.90** IS and **−12.34** FID at $1.6\times$ speedup.
>
> > **Experimental Designs Or Analyses**
> >
> - **The performance improvement of Learning-to-Cache is quite limited.** First, both methods without re-training the model gain comparable performance, which is also much better than that of the non-accelerated models in Tab. 13. Thus, it is reasonable to show our superiority with experiments at more challenging settings (i.e., higher CUR). For example, in Fig. 7, HarmoniCa can achieve substantially higher IS and lower FID scores, 30.90 and 12.34, under $1.6\times$speedup. Moreover, our method consumes less training time in these experiments, since we employ the same training settings (e.g., iterations and batch size) as Learning-to-Cache (analysis can be found in Sec. 4.3).
> - **The results in the tables do not match those in the original papers.** For the performance and speedup gaps of Learning-to-Cache between this work in Tab. 2 and the official paper, we have to clarify that we re-run [the official implementation of Learning-to-Cache with the provided arguments](https://github.com/horseee/learning-to-cache/tree/main/DiT) without any modification.  However, we conducted training procedures on 4 H800 GPUs and evaluation phases on 8 H800 GPUs, contrasting with the original experimental setup employing 8 A6000 GPUs for both stages.
> - **The results in Tab. 9 are inconsistent with Tab. 2.**  As mentioned in Sec. I, the implementation in the code of Learning-to-Cache is inconsistent with that in Alg. 1 in the official paper. The former only employs the caching strategy in non-adjacent steps, and the latter utilizes the caching strategy across all steps (the same as ours). The trick in the former can help mitigate the impact of accumulative error by the no-caching steps (4th row vs. 5th row in Tab. 15). To make a fair comparison, we implement Alg. 1 in the official paper and report the results in Tab. 9.

---

### Official Review · Reviewer_DPj7 · 2025-03-14

**Overall Recommendation:** 3

**Summary:**

This paper proposes HarmoniCa, a learning-based caching framework designed to accelerate Diffusion Transformersby reusing transformer block features during inference. HarmoniCa builds on the Learning-to-Cache framework by improving the training mechanism for the caching router. Extensive experiments demonstrate the effectiveness of this method in class-conditional image generation and text-to-image (T2I) generation.


## Post-Rebuttal Comments

I thank the authors for the rebuttal. While most of my initial concerns have been addressed, I note that two reviewers have raised valid points regarding the limited performance improvements over prior methods (e.g., CT5R, 5nDy). Given this, I maintain my rating.

**Claims And Evidence:**

Most claims in the paper are supported by empirical evidence, but some require further clarification:

1) **HarmoniCa accelerates Diffusion Transformers while maintaining high-generation quality.** Tables 1 and 2 show speedup improvements compared to the Learning-to-Cache framework and SA-Solver, demonstrating effectiveness in class-conditional image generation and text-to-image (T2I) tasks.

2) **Concerns regarding acceleration rates and resolution scaling.** Why do acceleration rates of Harmonica decline as resolution increases in PixArt-$\alpha$ and PixArt-$\Sigma$ (Table 2 [25 steps], and Table 12 in the supplementary material)? Is this due to memory bottlenecks associated with caching longer sequences?

**Essential References Not Discussed:**

I don't see this work missing any essential references.

**Experimental Designs Or Analyses:**

1) Compute cost for PixArt experiments: Can the authors provide the computational requirements for the PixArt experiments?
2) Absence of standard deviation in experimental results: While it may be computationally expensive to repeat all experiments, reporting standard deviations for at least one key setup would improve result reliability of the proposed caching method.

**Methods And Evaluation Criteria:**

1) **Sampler consistency during training and inference.** Should the sampler used during training match the one used at inference, given that the caching router is trajectory-dependent? If so, this introduces a limitation that should be clearly discussed.

2) **The evaluation metrics rely on FID for evaluating text-to-image generation which is concerning.** As FID relies on Inception model features trained on ImageNet-1K at 299×299 resolution, it may not be the best fit for evaluating PixArt models. Did the authors explore alternative evaluation metrics such as FID_{DINO} or human evals for a more comprehensive assessment?

**Other Comments Or Suggestions:**

N/A

**Other Strengths And Weaknesses:**

Please see the above sections for strengths and weaknesses: Claims And Evidence, Methods And Evaluation Criteria, and Experimental Designs Or Analyses.

**Questions For Authors:**

Please see the above sections for strengths and weaknesses: Claims And Evidence, Methods And Evaluation Criteria, and Experimental Designs Or Analyses.

Authors please consider addressing the weaknesses during rebuttal.

In my opinion, the weaknesses of this paper slightly outweigh the strengths. But I’m willing to change my opinion based on the rebuttal.

**Relation To Broader Scientific Literature:**

The paper builds on existing work in feature caching for diffusion models.

**Theoretical Claims:**

The paper does not introduce formal theoretical claims or proofs.

---

> ### Author Rebuttal · Authors · 2025-04-01
>
> Thanks for the reviewer’s constructive comments.
>
> > **Claims And Evidence**
> >
> - **Concerns regarding acceleration rates and resolution scaling.** Thanks for the insightful question. We agree that the reason why the acceleration rates of HarmoniCa decline ($1.67\times\rightarrow1.56\times$) with the resolution increase ($256\times256\rightarrow2048\times2048$) is due to memory bottlenecks. Specifically, HarmoniCa achieves similar CUR (i.e., cache usage ratio) for these resolutions, but requires a cache size increase from 936 MB (BS=8) to 4320 MB (BS=1). Thus, we believe preloading and asynchronous saving can help resolve the problem and will explore this in the future.
>
> > **Methods And Evaluation Criteria**
> >
> - **Sampler consistency during training and inference.**  The sampler used during training is not required to match the one used at inference. We have already included the discussion in Sec.  K. However, mismatched samplers can lead to a discrepancy in sampling trajectories. This does affect performance as reflected in our experiments. Therefore, we suggest that the sampler should be the same for optimal performance.
> - **The evaluation metrics rely on FID for text-to-image generation.**  We have conducted experiments with more metrics (e.g., Image-Reward, LPIPS, and PSNR), which are provided in the Appendix (Sec. J). Here, we also consider DINO and human evaluations (e.g., [HPSv2](https://arxiv.org/pdf/2306.09341) and [PickScore](https://arxiv.org/pdf/2305.01569)) as suggested. As shown in the following tables, our method with a higher speedup ratio far outperforms baselines and achieves comparable performance with non-accelerated models.
>
>     *We use the configurations and models in Tab. 2 in the manuscript to generate images for evaluation.*
>
>     | Model | DINO $\uparrow$ | HPSv2 $\uparrow$ | PickScore $\uparrow$ | Latency (s)$\downarrow$ | Speedup$\uparrow$ |
>     | --- | --- | --- | --- | --- | --- |
>     | PixArt-$\alpha$ $256\times256$ | 0.3082 | 28.91 | 27.89 | 0.553 | - |
>     | + 15 steps | 0.2582 | 27.98 | 23.02 | 0.418 | $1.32\times$ |
>     | + FORA | 0.2712 | 28.11 | 22.44 | 0.364 | $1.52\times$ |
>     | + HarmoniCa | **0.3235** | **28.72** | **26.65** | **0.346** | **$1.60\times$** |
>     | PixArt-$\alpha$ $512\times512$ | 0.3339 | 30.53 | 28.52 | 1.759 | - |
>     | + 15 steps | 0.3127 | 29.79 | 22.03 | 1.291 | $1.36\times$ |
>     | + FORA | 0.3099 | 29.82 | 21.98 | 1.150 | $1.53\times$ |
>     | + HarmoniCa | **0.3289** | **30.28** | **27.47** | **1.072** | $1.64\times$ |
>
> > **Experimental Designs Or Analyses:**
> >
> - **Compute cost for PixArt experiments.** Thank you for the question. We provide the computation costs of training and inference in the following table. It is worth noting that we only use naive DDP training without any advanced strategies like gradient checkpointing and FSDP. Thus, the requirements of computation can be further decreased in practice.
>
>     *The settings of training can be found in Sec. 5. We employ BS=1 here to obtain the memory consumption of inference.*
>
>     | Model | Infer. Mem. (GB/GPU)$\downarrow$ | Train. Mem. (GB/GPU)$\downarrow$ | Train. Time (h)$\downarrow$ |
>     | --- | --- | --- | --- |
>     | PixArt-$\alpha$ $256\times256$ | 20.86 | - | - |
>     | + HarmoniCa | 21.21 | 79.05 | 1.84 |
>     | PixArt-$\alpha$ $512\times512$ | 24.11 | - | - |
>     | + HarmoniCa | 24.61 | 77.30 | 2.92 |
> - **Absence of standard deviation in experimental results.** Thank you for the suggestion.  We conducted five independent runs with different random seeds on the following setting: PixArt-$\alpha$ $256\times256$, using DPM-Solver++ (20 steps) and evaluating on 5,000 images. The results show high consistency across runs, and more importantly, our caching-accelerated models consistently outperform the non-accelerated models in all evaluation metrics, confirming our method's robustness and effectiveness.
>
>     *”x/y’’ in the table denotes the results that come from non-accelerated models and HarmoniCa. We use the same settings in Tab. 2 in the manuscript.*
>
>     | Seed | IS$\uparrow$ | FID$\downarrow$ | sFID$\downarrow$ |
>     | --- | --- | --- | --- |
>     | 8 | 33.28/33.27 | 37.31/35.44 | 94.78/92.10 |
>     | 16 | 33.61/33.62 | 37.43/35.46 | 94.74/92.11 |
>     | 24 | 33.59/33.60 | 37.55/35.48 | 95.01/92.32 |
>     | 32 | 33.65/33.65 | 37.11/35.34 | 94.87/92.13 |
>     | 40 | 33.64/33.64 | 37.05/35.29 | 94.88/92.19 |
>     | deviation | 0.138/0.144 | 0.188/0.073 | 0.093/0.081 |

---

> > ### Comment · Reviewer_DPj7 · 2025-04-03
> >
> > Thank you for the response. The rebuttal addresses my concerns, and I don’t have any additional requests. I’ve also read the other reviews and responses, and I see that there are concerns regarding the technical novelty (5nDy, CT5R). Given that, I’d like to keep my current recommendation.

---

> > > ### Author Response · Authors · 2025-04-03
> > >
> > > We sincerely appreciate the reviewer’s time and valuable feedback.
> > >
> > > Additionally, we wish to highlight that the proposed method brings significant technical novelty, which has been justified by our response of **Novelty Concerns** (1st round rebuttal) and **Performance Significance** (2nd rebuttal) to 5nDy. We kindly request the reviewer to re-evaluate the technical merit of our work in light of these sufficient justifications. Should any specific concerns remain, we would be grateful for the opportunity to provide additional clarification.

---

### Decision · Program_Chairs · 2025-05-01

**Decision:**

Accept (poster)

**Comment:**

The paper received three reviews that positively assessed the work and recommended acceptance. The fourth review recommended a strictly negative rating. During the rebuttal period, there was healthy engagement between the authors and the reviewers, during which many concerns were addressed, and the scores either remained the same or improved. However, the authors were not able to change the opinion of Reviewer 5nDy. They composed a message to the AC in which they outlined their disagreements with Reviewer 5nDy. While the AC believes that certain statements made by 5nDy have merit, the AC feels that their score may be overly critical of the work. The other reviewers also raised concerns regarding differences with L2C; however, they accepted the arguments provided by the authors.

Hence, the AC believes that the present paper is a good piece of work and recommends acceptance.